



# Diurnal cycle of coastal anthropogenic pollutant transport over southern West Africa during the DACCIWA campaign

Adrien Deroubaix[1,2], Laurent Menut[1], Cyrille Flamant[2], Joel Brito[9], Cyrielle Denjean[6], Volker Dreiling[7], Andreas Fink[5], Corinne Jambert[4], Norbert Kalthoff[5], Peter Knippertz[5], Russ Ladkin[8], Sylvain Mailler[1], Marlon Maranan[5], Federica Pacifico[4], Bruno Piguet[6], Guillaume Siour[3], and Solène Turquety[1]

[1]LMD/IPSL, École Polytechnique, Université Paris Saclay, ENS, IPSL Research University; Sorbonne Universités, UPMC Univ Paris 06, CNRS, Palaiseau, France
[2]LATMOS/IPSL, Sorbonne Université, Université Paris-Saclay & CNRS, Paris, France
[3]LISA/IPSL, Laboratoire Interuniversitaire des Systèmes Atmosphériques (LISA), UMR CNRS 7583, Université Paris Est Créteil et Université Paris Diderot, Institut Pierre Simon Laplace, Créteil, France
[4]LA, Laboratoire d Aérologie, University of Toulouse, CNRS, UPS, Toulouse, France.
[5]KIT, Institute of Meteorology and Climate Research, Karlsruhe Institute of Technology, Germany
[6]CNRM, Centre National de la Recherche Météorologique, UMR3589, CNRS, Météo-France, Toulouse, France
[7]DLR, Deutsches Zentrum für Luft-und Raumfahrt, Oberpfaffenhofen, Germany
[8]BAS, British Antarctic Survey, Cambridge, UK
[9]LAMP, Laboratoire de Météorologie Physique, Université Clermont Auvergne, Aubière, France

*Correspondence to:* Adrien.Deroubaix@lmd.polytechnique.fr

**Abstract.** During the monsoon season, pollutants emitted by large coastal cities and biomass burning plumes originating from Central Africa have complex transport pathways over Southern West Africa (SWA). The Dynamics-Aerosol-Chemistry-Cloud-Interactions in West Africa (DACCIWA) field campaign has provided numerous dynamical and chemical measurements in and around the super site of Savè in Benin ($\approx$ 185 km away from the coast), which allows quantifying the relative contribution

5 of advected pollutants. Through the combination of in-situ ground measurements with aircraft, radio-sounding, satellite and high-resolution chemistry-transport modeling with the CHIMERE model, the source attribution and transport pathways of pollutants inland (here, $NO_x$ and CO) are carefully analyzed for the 01 – 07 July 2016 period. The relative contributions of different sources (*i.e.* emissions from several large coastal cities) on the air quality in Savè are characterized. It is shown that a systematic diurnal cycle exists with high surface concentrations of pollutants from 18:00 to 22:00 UTC. This evening peak

10 is attributed to pollution transport from the coastal city of Cotonou (Benin). Numerical model experiments indicates that the anthropogenic pollutants are accumulated during the day close to the coast, and transported northward as soon as the daytime convection in the atmospheric boundary layer ceases after 16:00 UTC, reaching 8°N at 21:00 UTC. When significant biomass burning pollutants are transported into continental SWA, they are mixed with anthropogenic pollutants along the coast during the day, and this mixture is then transported northward. At night, most of the coastal anthropogenic plumes are transported

15 within the planetary boundary layer (below about 500 m above ground level), whereas the biomass burning pollutants are mostly transported above it, thus generally not impacting ground level air quality.



# 1 Introduction

The United Nations Department of Economic and Social Affairs, Population Division reported 31 megacities globally (urban agglomerations with more than 10 million inhabitants in 2016) and that their number is projected to rise up to 41 by 2030. In Southern West Africa (SWA), Lagos is considered as a megacity (with more than 13 million inhabitants) and is expected to reach 24 million in 2030. The urban agglomeration extents along the coast to Cotonou (Benin) and even to Lomé (Togo). Moreover Accra in Ghana (with a population predicted to increase from 2.3 in 2016 to 3.3 million in 2030), Kumasi in Ghana (with a population predicted to increase from 2.7 in 2016 to 4.2 million in 2030) and Abidjan in Ivory Coast (with a population predicted to increase from 5.0 in 2016 to 7.8 million in 2030) will all contribute to form a more or less continuously urbanized strip at some point during the 21th century. This growth is associated with enhanced pollutant emissions and low air quality, which leads to chronic health problems (Lelieveld et al., 2015) and contributes to anthropogenically forced climate change.

The vertical structure of air pollution is complex along the Guinea Coast during the period when the West African Monsoon (WAM) is established in the boreal summer. From the surface to the top of the Planetary Boundary Layer (PBL), marine air transported by the northeastward monsoon flow gets enriched with anthropogenic pollution emitted at the coast before moving further inland (Knippertz et al., 2015b). Above the marine PBL, biomass burning aerosol layers, resulting from incomplete combustion of fires in Central Africa (Giglio et al., 2006; Zuidema et al., 2016) can be observed on occasion reaching the Guinean coast after being transported over thousands of kilometers (Menut et al., 2018). In higher layers, at altitudes from 3 to 5 km, the Saharan Air Layer (SAL) is generally observed to be advected from the north depending on the meridional disturbances of African Easterly Jet (AEJ), carrying desert dust (Flamant et al., 2009; Lafore et al., 2011; Crumeyrolle et al., 2011). This general picture is often perturbed by the presence of organized convective systems, which propagate along the Guinea Coast from Nigeria to Liberia (Maranan et al., 2018). The latter authors also note the presence of land-sea breeze convective systems in the immediate coastal strip.

The EU-funded project Dynamics-Aerosol-Chemistry-Cloud-Interactions in West Africa (DACCIWA) was designed to focus specifically on the Guinea Coastal atmospheric dynamics and the interactions between aerosols, chemistry and clouds (Knippertz et al., 2015a). An intensive measurement campaign took place in Nigeria, Benin, Togo, Ghana and Ivory Coast during June-July 2016, which corresponds to the climatological onset period of the WAM (Janicot et al., 2008). Three research aircraft flew over the Guinea Coastal region with different scientific objectives, notably with flight plans designed to map out city, shipping and flaring emissions or focused on sampling biomass burning aerosols layers (Flamant et al., 2018b). The DACCWIA field campaign took place in so-called post WAM onset conditions, *i.e.* after deep convection (and related precipitation) had migrated from the coast, inland over the Sahel (Knippertz et al., 2017).

During the WAM, the atmospheric composition over the Gulf of Guinea coastal region is the result of a complex mix of natural and anthropogenic sources, which include urban, biomass burning, biogenic, desert dust and oceanic compounds. Using numerical tracer experiments Menut et al. (2018) have highlighted that fire emissions in Central Africa impacting the surface aerosol and gaseous species concentrations over the Gulf of Guinea are mostly transported over the southeast Atlantic above the marine PBL. Using WRF-CHIMERE numerical simulations of the WAM during the African Monsoon





Multidisciplinary Analyses (AMMA) campaign (May–July 2006), Deroubaix et al. (2018) quantified the relative contribution of anthropogenic and biomass burning sources to carbon monoxide (CO) concentrations over SWA, which in July is about 25 % local anthropogenic and 50 % biomass burning from Central Africa. The remaining 25 % are the background corresponding to long-range transport from outside of Africa. However the high biomass burning contribution is partly due to the significant

under-estimation anthropogenic emission for the Gulf of Guinea region (Marais et al., 2014; Liousse et al., 2017; Keita et al., 2018).

Over SWA, Adler et al. (2017) and Deetz et al. (2018) have documented a regular occurrence of a coastal front, which is located where the strongest horizontal gradients of wind speed and potential temperature occur. It develops during daytime and propagates inland in the evening. After the frontal passage, the wind in the lowermost troposphere brings air masses from

the coast northward, especially at night with the Nocturnal Low-Level Jet (NLLJ) (Schuster et al., 2013), and probably also anthropogenic pollutants emitted from coastal urban areas (e.g. Djossou et al., 2018) and biomass burning pollutants imported by monsoon flow (Haslett et al., in preparation).

The main objective of this article is to understand the diurnal cycle of anthropogenic pollutant transport from the coast to the continental SWA. We present numerical tracer experiments made with high-resolution CHIMERE simulations set up in order

to separate the contribution of each important urban agglomeration, namely Abidjan, Accra, Lomé, Cotonou and Lagos. We take advantage of the DACCIWA measurements made by the three research aircraft, by an enhanced radio-sounding network, and at the super-site of Savè in central Benin (Knippertz et al., 2015a). The super-site of Savè serves as a representative location of inland air quality. We focus on the period 1–7 July, during which a case of long-range transport of biomass burning aerosol from Central Africa was observed (Flamant et al., 2018b). We aim at answering the following questions:

– *What is the relative contribution of each coastal urban area to the air pollution at Savè? How does it evolve during the day?*

– *How are biomass burning and anthropogenic pollutants mixed along the coast and inland? Is it usually a mixture of the two pollutions that is transported inland in the PBL?*

We shall answer these questions using a synergistic combination of observations and numerical modeling experiments,
described in Section 2. Section 3 analyzes the temporal evolution of meteorology and air pollution over a portion of SWA including Ivory Coast, Ghana, Togo, Benin and Nigeria. Then, we focus on Urban Anthropogenic (URB) and long-range Biomass Burning (BB) pollutant transport in Section 4. Conclusions are given in Section 5.

## 2   DACCIWA project: observations and modeling

In the DACCIWA project, there are strong components on both in-situ observations and modeling. Here, we present all studied
sites (2.1), observational datasets used (section 2.2), and numerical simulations performed to analyze the pollution transport pathways (section 2.3).



## 2.1 Studied sites

We focus on six locations, five major urban agglomeration of the Guinean Coastal region, and one small town, Savè, which is 185 km north of Cotonou (Benin). Table 1 shows the coordinates and the population of the urban agglomerations studied. For Abidjan, Accra, Lomé and Lagos, we present estimations of the department of social and economic affairs of United Nations.

Abidjan, Accra and Lomé are associated with large administrative areas. The population of the Lagos urban agglomeration is controversial because the last census was in 2006 (en.wikipedia.org/wiki/Lagos details therein).

  Comparing Lomé and Cotonou, Lomé is a large administrative area in Togo while Cotonou is a city with very high density over a small area. According to General Population and Habitat Census of Togo (2014, www.stat-togo.org/), the population of Lomé is about 837,000 inhabitants in the city, and is about 1,570,000 inhabitants in the administrative state. Cotonou city is

estimated by the World Urbanization Prospects (The 2011 Revision) to have about 1,086,000 inhabitants.

  According to General Population and Habitat Census of Benin (2013, www.insae-bj.org/recensement-population.html), the population of Cotonou only slightly increased by 2.09 % over the period 2002-2013 because of the limited possibility of expansion. They note that along the shores of Lake Nokoué the population has increased rapidly, thus forming an agglomeration of 2,194,000 inhabitants, calculated as the sum of the Cotonou district and several cities of the Atlantique district (Abomey

Calavi and So-Ava) and of the Ouémé district (Seme Kpodji, Porto Novo, Avrankou and Akpro-Misserete).

## 2.2 Observational datasets

During the DACCIWA field campaign, several observational platforms were deployed to perform in-situ and remote sensing measurements (Knippertz et al., 2017; Flamant et al., 2018b). In this study, we use datasets acquiered by ground based stations, aircrafts, radiosondes and satellites. Table 2 gives the main information on each dataset. Figure 1 presents the location of the

aircraft flight tracks, and of the stations.

    – Ground based station

  At Savè, the Karlsruhe Institute of Technology (KIT) and the Paul Sabatier University (UPS) have set-up one of the three DACCIWA super-sites. UPS installed a chemical analyzer (ThermoEnvironment Instrument) which measured $NO_2$, NO and CO surface concentrations (Pacifico et al., 2018). Raw observations acquired at 10 s are averaged hourly. The detection limit

of the instrument is 0.05 ppb for $NO_2$ and NO, and 12 ppb for CO (Derrien and Bezombes, 2016). The measurements site is upwind of the Savè city when the wind corresponds to the monsoon flow (SW sector). On 3 July from 18:00 to 21:00 UTC, the wind direction was shifted, which corresponds to local pollution. This period has been removed from the analysis.

    – Aircraft campaign

  The DACCIWA aircraft campaign took place during the period 25 June - 14 July 2016 and was based at the Lomé (Togo)

airport (Flamant et al., 2018b). Three research aircraft were involved: a Twin Otter operated by the British Antarctic Survey (BAS), an ATR-42 operated by the French "Service des Avions Français Instrumentés pour la Recherche en Environment"





(SAFIRE), and a Falcon operated by the German "Deutsches Zentrum für Luft und Raummfahrt" (DLR). We base our study on three variables, namely relative humidity, wind direction and wind speed, which are measured by core meteorological instrumentation. The flights trajectories used are depicted in Figure 1. For the three aircraft, raw observations acquired at 1 Hz are averaged over 3-minute time steps.

– Radiosonde campaign

The DACCIWA project included a large radiosonde component with locations carefully chosen building on the AMMA radiosonde campaign experiences (Parker et al., 2008; Lothon et al., 2008; Fink et al., 2011; Schuster et al., 2013). We use radiosondes launched from four locations: Abidjan, Accra, Cotonou and Savè (*cf.* Table 1). There were four releases per day at around 00:00 UTC, 06:00 UTC, 12:00 UTC and 18:00 UTC. In Savè, more radiosondes were launched at the super-site during

the Intensive Observation Period of 1-7 July 2016, (Kalthoff et al., 2018).

     – MODIS sensor

We analyze the spatial extent of BB layers from satellite observations of Aerosol Optical Depth (AOD) at 550 nm made by MODIS (MODerate-resolution Imaging Spectroradiometer) on both the Aqua platform (MYD08-D3-6 dataset, passing over the studied region at 13:30 UTC) and the Terra platform (MOD08-D3-6 dataset, passing over the studied region at 10:30 UTC).

Daily MODIS AOD averages at $1°$ resolution have been calculated from the Collection 6 combined product of the Dark-Target retrieval available over oceans or non-bright continental surface, and the Deep-Blue retrieval available over deserts (Sayer et al., 2013; Hsu et al., 2013; Sayer et al., 2014).

     – CALIOP sensor

In order to identify the altitude of aerosols with their speciation, we use the space-borne Cloud-Aerosol Lidar with Or-

thogonal Polarization (CALIOP) with aerosol types classification (Winker et al., 2009). The CALIOP cross-sections are very useful since this is a realistic way to have an instantaneous evaluation of the aerosol layer altitudes, with their type depending on backscattered optical measurements, *e.g.* Winker et al. (2013). Data are available on https://www-calipso.larc.nasa.gov/ products/lidar/browse_images.

## 2.3   Numerical modeling by WRF-CHIMERE models

The WRF-CHIMERE simulations presented in this study have a similar set-up to the one used by Deroubaix et al. (2018). Both models are run offline in a nested configuration on the same grids. Two nested domains are used over the period 1 to 7 July: a regional domain (10 km×10 km) and a high-resolution coastal domain (2 km×2 km). In the following, we present only results modeled over the high-resolution domain. The studied period (1 to 7 July) is entirely included in a WAM post-onset phase, which has been defined from 22 June to 20 July by Knippertz et al. (2017).



### 2.3.1 Meteorological fields from the WRF model

Meteorological variables are modeled with the regional non-hydrostatic WRF model (version 3.7.1, Skamarock and Klemp, 2008). The domains have a constant horizontal resolution with 32 vertical levels from the surface to 50 hPa. We use a 2-way nesting for the communication between different domains.

Global meteorological fields are taken from the US Global Forecast System (operational final analyses) produced by the National Center for Environmental Prediction. These fields are used to nudge hourly fields of pressure, temperature, humidity and wind in the WRF simulations, with spectral nudging, which has been evaluated for regional models by von Storch et al. (2000). In order to enable the PBL variability to be resolved by WRF, low frequency spectral nudging is used only above 850 hPa.

The WRF model set-up is as follows: The microphysics scheme is the Single Moment-6 class microphysics scheme (WSM6), the radiation scheme is the Rapid Radiative Transfer Model for General Circulation Models (RRTMG) with the Monte-Carlo Independent Column Approximation (McICA) method of random cloud overlap from Mlawer et al. (1997), the PBL physics are computed using the Yonsei University scheme (Hong et al., 2006), the cumulus parametrization is the ensemble Grell-Dévényi scheme, the surface layer scheme is the Carlson-Boland viscous sub-layer and the surface physics is calculated with

the 'Noah' Land Surface Model scheme with four soil temperatures and moisture layers (Ek et al., 2003). This set-up has already been used by Deroubaix et al. (2018) because it allows to reproduce a satisfactory diurnal cycle of wind speed over SWA according to Schuster et al. (2013).

### 2.3.2 Gaseous tracers transport from the CHIMERE model

CHIMERE is a regional chemistry-transport model (version 2017), fully described in Menut et al. (2013) and Mailler et al.

(2016). The 32 vertical levels of the WRF model are projected onto the 20 levels for CHIMERE from the surface and up to 200 hPa.

In this study, the model is used in its tracer version and there is no atmospheric chemistry. Since we want to distinguish the relative contribution of the pollution of several cities inland, we designed a first experiment for which we impose pseudo-anthropogenic emissions at specific locations: Abidjan (Ivory Coast), Accra (Ghana), Lomé (Togo), Cotonou (Benin), Lagos

(Nigeria) (*cf.* Table 1 for coordinates). Specific tracers are emitted for a given city in order to distinguish their relative contributions.

The city emissions are constant and continuous during the modeled period (1–7 July). This allows quantifying the variability due to the meteorology only. The tracer emissions occur in the grid cell of each city. Emissions are released at the lowest level of the model (below 10 m altitude). Emissions are proportional to the population of each city. We defined an arbitrary emission

for one million inhabitants. Then we multiply this emission by a factor depending on the population (*cf.* Table 1): 1.8 for Lomé, 2.2 for Cotonou, 3 for Accra, 5 for Abidjan and 13 for Lagos.

A second objective is to understand the interactions between URB and BB resulting from long-rang transport on 5 July. For this, we design a second numerical tracer experiment in which BB tracers are added to the URB tracers, and are tagged to be





different from the URB tracers. BB tracers are released to reproduce the BB layer observed with MODIS on 5 July at 00:00 UTC to 6 July at 23:00 UTC with a spatial horizontal extent going from 1°W to 2°E at 4.5°N, and at an altitude of ≈ 1.5 km (*cf.* Figure 9, and Sections 3.1 and 4.3 for justifications).

The two tracer experiments are summarized in Table 3. They are accessible on the DACCIWA database (http://baobab.sedoo.
fr/Data-Search/?datsId=1760&project_name=DACCIWA).

The tracers are transported using the van Leer scheme (van Leer, 1979). There is no sink for tracers (no deposition and no chemical reaction). The tracers are chosen to be gaseous and are representative of the gaseous part of the URB emissions and the BB plume. This choice of gaseous tracers was made to be consistent with the gaseous concentrations measured by the aircraft and the surface gaseous concentrations measured at the Savè super-site. The only difference to aerosol is the absence
of settling. But this long-term impact could be considered as negligible in this study focused on a few days and a spatially restricted region: it is assumed that gaseous and aerosol species are transported in the same way by the meteorological flow.

## 3 Large scale atmospheric patterns over the Gulf of Guinea

This section is dedicated to analysis of the atmospheric dynamics, thermodynamics and composition across SWA using AOD from satellites in Section 3.1, together with relative humidity and wind from radiosondes in Section 3.2 and from aircraft
in Section 3.3. The prerequisite to realistic numerical tracer experiments is the accuracy of the meteorological simulation. The WRF meteorological simulation is therefore extensively compared to in-situ observations made by both radiosondes and aircrafts.

### 3.1 Regional scale aerosol distribution

This section investigates the daily MODIS-AOD observations for the period 1 to 7 July 2016. Two important types of aerosols
can be advected towards SWA: Dust from the north and BB from the south. We focus on two different days, 3 and 5 July 2016 (Figures 2 and 11). Note that we present one day moving averages (Figures 2) because we analyze the long-range transport of aerosols (using MODIS level-3 product with a coarse resolution of 1°).

During the studied period, high AOD values are found north of the domain over the Sahel (north of 14°N) and south of the domain over the Gulf of Guinea (on average over the period 1–7 July, Figure 11). The origin of the high AOD over the Gulf of
Guinea is well known. This is the BB layer coming from Central Africa, where intense vegetation fires occur during this season (Giglio et al., 2006). A part of this pollution is transported over the Gulf of Guinea and the BB plume reaches the Guinea Coast depending on the synoptic wind patterns (Menut et al., 2018). It is worth noting that during this period, there is no evidence of mineral dust transport over the studied cities.

3 and 5 July 2016 are two contrasting days in terms of AOD values over the Gulf of Guinea (Figure 2). On 2 July, AOD
30 values are low over the continent and moderate over the ocean (Figure 11), which is in agreement with Flamant et al. (2018a) who have shown that the BB layer is present close to the Guinean coast but it is not reaching the coast. On 3 July, AOD values are low to moderate over our domain (AOD < 0.5, Figure 2). On 4 July, there is a pattern of high AOD (AOD > 0.5) 100 km





south of the coast (Figure 11). On 5 July, the BB layer reaches the coastline (Figure 2), then on 6 July it seems to penetrate inland but clouds prevent AOD retrievals over Togo and Benin (Figure 11). On 7 July, this layer is no longer visible close to the Guinea Coast (Figure 11).

For 6 July, Flamant et al. (2018b) have shown a clear large-scale BB signature between Abidjan and Accra with in-situ
measurements made onboard the research aircraft. Moreover Brito et al. (2018) have analyzed atmospheric chemistry and demonstrated mixing of urban pollutants with advected BB into the region. This interpretation has been supported by backward trajectories locating the origin of the BB plume in Central Africa. The presence of this layer is also confirmed over the Gulf of Guinea using CALIPSO data acquired on 5 July (Figure 12), which gives a layer altitude between 1 and 3 km above mean sea level (amsl).

## 3.2   Vertical layer identification

In this section, we combine observations from the high-resolution radiosondes (Section 3.2.1) and the three aircraft (Section 3.2.2) over the period 1–7 July. For radiosondes, we analyze 32 vertical profiles in Abidjan, 32 in Accra, 26 in Cotonou and 51 in Savè (Figure 3). For aircraft data, we analyzed eleven flights including six of the ATR-42, four of the Falcon and one of the Twin Otter (Table 4). Aircraft observations are acquired only during daytime. The modeled and observed dynamical and
thermodynamical variables are compared in order to identify the different layers. The modeled variables have been interpolated along the balloon trajectories and aircraft flight tracks using a spatial bilinear interpolation, and then temporal and vertical linear interpolations with a 3-minute time step.

### 3.2.1   From radiosondes

For wind speed, the vertical profiles observed at the four locations have a similar shape (Figure 3). The mean wind speed
increases from the surface to 300 m amsl, then decreases to 3 m.s$^{-1}$ at 1.5 km amsl and finally increases to a maximum of about 8 m.s$^{-1}$ between 3 and 5 km amsl. The model predicts a vertical wind profile in good agreement with observations from the surface to 300 m amsl but there is an increasing positive bias from 300 m to 1 km amsl, reaching +2 m.s$^{-1}$ at 1 km amsl. At 300 m amsl, observed wind speed reaches 6 m.s$^{-1}$ on average, which shows the NLLJ signature (Schuster et al., 2013). The model reproduces this signature with wind speed reaching 7 m.s$^{-1}$ but over-estimates its altitude (at 400 m amsl) at the three
coastal sites (Abidjan, Accra, Cotonou).

The vertical profile observed at Savè stands out from the three other cities with lower wind speed below 1 km and higher wind speed above 2 km amsl. The lower wind speed near the surface may be related to the greater distance from the coast resulting in a stronger deceleration by friction, which is not reproduced by the model. When looking at 3 km amsl, this is close to the altitudinal maximum of the AEJ. Savè is located at a latitude closer to the AEJ core, which is seen at about 10°N
(Knippertz et al., 2017). The jet is clearly observed only at Savè, with wind speeds up to 10 m.s$^{-1}$ at 3 km amsl, which is modeled in good agreement.

For wind direction, the four cities have again similar profiles. The mean observed and modeled vertical profiles are composed of three distinct layers. From the surface to 1 km amsl, the monsoon layer corresponds to wind coming from the sector between





210° and 240°. From 2 to 5 km amsl, wind direction is also almost constant between 80° and 120°. In between these two layers, which are well defined in terms of direction, there is a layer characterized by a quick change of direction from 240° to 120°. This layer associated with weak wind speed is a directional shear layer. On average, the monsoon layer depth seems to be over-estimated by about 200 m because the modeled wind direction is biased by about +20° between 1.2 and 2 km amsl.

5 For the three variables, the profiles of their standard deviation present the same modeled and observe characteristics (gray shading in Figure 3). For wind speed, the standard deviation is about 2 m.s$^{-1}$ from the surface to 2 km amsl, and it increases up to 4 m.s$^{-1}$ from 2 to 4 km amsl. For wind direction, the standard deviation is low (about 45°) from the surface to 1 km amsl, it increases from 1 to 2 km in the directional shear layer and it decreases from 2 to 4 km amsl. For RH, the standard deviation is about 10 % from the surface to 2 km amsl, and it increases in the AEJ layer but the model does not reproduce the low RH 10 observed values in this layer.

This analysis shows that the modeled monsoon layer is too deep when it arrives at the coast. Further inland the monsoon flow is too fast when it reaches Savè. The comparison between observed and modeled meteorology also reveals that the model reproduces well enough the several vertical layers in terms of wind direction and speed, and thus most likely the transport that we want to characterize using the tracer experiments.

### 15 3.2.2 From research aircraft

The atmospheric vertical structure can be separated into three different layers (*cf.* Section 3.2): (i) from the surface to 1 km amsl, there is the monsoon flow with RH > 90 %, wind speed > 4 m.s$^{-1}$ and direction from the southwesterly sector; (ii) between 1 to 2 km amsl, there is a directional shear layer associated with low wind speed and changing wind direction; (iii) between 2 and 4 km amsl, this is the AEJ layer with RH < 80 %, reversed wind direction coming from the northeast and 20 wind speed up to 8 m.s$^{-1}$. In order to evaluate the model, aircraft measurements during daytime are separated into three corresponding altitude ranges (Table 4).

In the directional shear layer, from 1 to 2 km amsl, modeled wind direction is in good agreement with observations (relative bias lower than 1 %), although it is associated with a wider distribution (observed inter-quartile range $191.17 - 285.29$ ° and modeled $214.17 - 237.67°$) than in the first kilometer (observed inter-quartile range $220.08 - 244.41°$ and modeled $207.29 -$ 25 $272.28°$). The modeled and observed wind speed distributions are narrower than in the monsoon layer (observed inter-quartile range $2.10 - 4.04$ m.s$^{-1}$ and modeled $1.88 - 3.94$ m.s$^{-1}$), showing that this layer is well defined over the domain.

In the AEJ layer, from 2 to 4 km amsl, the observed distribution of wind speed is wider than at lower altitudes, which is in good agreement with the modeled distribution. Observed and modeled RH and wind direction distributions are also consistent.

From the surface to 1 km amsl, the modeled wind speed and direction match well the observations (absolute bias lower than 30 0.2 m.s$^{-1}$ and 4°, respectively). The observed distribution of the monsoon wind is captured by the model up to 1 km amsl (observed inter-quartile range $3.80 - 6.47$ m.s$^{-1}$ and modeled $3.98 - 6.80$ m.s$^{-1}$). The modeled distribution of RH shows a dry bias of the model in the monsoon flow (absolute bias of -6 %).

During daytime, the monsoon layer is reproduced with a dry bias and with a low (relative) bias of wind speed (+4 %) and direction (-2 %), which is of prime importance to accurately model the URB transport.



### 3.3 From the coast to the North: 5 July Lomé-Savè flights

As we want to understand northward pollution transport, we need to focus on the wind direction and speed from the coast to the north. In this section, we analyze the spatial variability of the wind over the Lomé–Savè transect. We compare aircraft measurements of wind to modeled values using data acquired during three specific flights conducted on 5 July at different

times of day with similar flight plans (*cf.* Figure 1) and similar altitude ranges (*i.e.* flying mostly below 2 km amsl). The French ATR-42 flight took place between 08:00 and 11:00 UTC, that of the German Falcon between 11:20 and 15:00 UTC, and that of the British Twin Otter between 16:00 and 17:50 UTC.

Using ceilometer measurements, Flamant et al. (2018b) have described the cloud base height evolution on 5 July at Savè, which was between 200 and 1000 m during the ATR-42 fight, between 400 and 1800 m during the Falcon flight, and between

1000 and 3800 m during the Twin Otter flight. This was a cloudy day, which allowed the operational center to plan for characterizing the diurnal cycle of low level clouds (Flamant et al., 2018b).

During the ATR-42 flight in the morning, measurements of wind speed range from 2 to 10 m.s$^{-1}$ (Figure 4). The highest values are observed close to the coast (greater than 8 m.s$^{-1}$) during the two times the aircraft passes there. The model reproduces the spread of the observed values below 2 km amsl. When the altitude reaches 2 km amsl, observed wind speed decreases

below 4 m.s$^{-1}$. Wind direction ranges from 240° to 300°, even at about 2 km amsl. The model predicts a constant direction at 250°, except when flying above 2 km amsl because the modeled direction changes, revealing an under-estimation of the modeled PBL depth.

In the morning, the monsoon layer is modeled with an over-estimation of the wind speed and with a sharp directional shear at 2 km amsl, whereas we observe an important variability of wind speed without reaching the directional shear layer up to 2

km amsl. This behavior of the model suggests that low level clouds are not well represented, leading to modeled PBL depth under-estimation.

During the Falcon flight around midday, wind speed also decreases from the beginning of the flight at the coast (up to 10 m.s$^{-1}$) to 100 km further North (less than 4 m.s$^{-1}$). Wind direction varies smoothly from 250° at the coast to 300° close to Savè. The model is able to reproduce the weakening of the monsoon layer linked to daytime dry convection (Adler et al., 2017;

Deetz et al., 2018), and the variability of observed and modeled wind direction and speed are in better agreement.

In contrast to the ATR-42 and Falcon flights, the Twin Otter flew only one time over Savè and made three vertical profiles up to 3 km amsl. During the Twin Otter flight in the afternoon, the range of wind speed increases compared to the two previous flights, reaching between 1 and 12 m.s$^{-1}$. The model reproduces well the wind in the monsoon layer but does not capture the wind direction changes between 1 and 3 km amsl. There is no clear change of the direction during the first sounding, but during

the latter two soundings, wind direction changes from southwesterly winds at 1 km amsl (about 225°) to northeasterly winds at 3 km amsl (about 45°). The model predicts a too sharp wind direction change from 1 to 3 km amsl, which shows that the directional shear layer depth is under-estimated.

Overall below 1 km amsl, wind speed ranges from 4 to 10 m.s$^{-1}$ with a direction from about 250°, which is in good agreement with the model. Knippertz et al. (2017) have shown that 5 July was during a period when the AEJ weakens and





becomes more fragmented, which has led to relatively patchy signals in wind and vorticity (described as feature G in this study). This results in observed wind direction mostly greater than the third quartile of the distribution measured over the period 1–7 July 2016 (Q3=244° below 1 km amsl, *cf.* Table 4).

The three aircraft cover the same region from 08:00 to 17:00 UTC. It is thus possible to quantify the evolution of dynamical variables during daytime. We have selected a box crossed many times by the aircraft in order to compare hourly averages of in-situ wind speed and direction observations to the modeled values. The box is delimited in latitude from 6.6 to 7.8°N, in longitude from 1.5 to 2.2°E and in altitude from 300 to 1000 m amsl (Figure 13). When the three flights cross this box, the average and standard deviation are calculated from observed values. For the model, we present the average and standard deviation of each hour calculated from all grid cells included in the box.

Wind speed observations decrease from 08:00 to 13:00 UTC (Figure 13), then increase again in the afternoon (but there are only few measurements made by the Twin Otter aircraft in the two boxes at 17:00 UTC). The model does not capture well the morning evolution, when the NLLJ is eroded. We note that the minimum of wind speed is modeled and observed in the early afternoon, when vertical mixing is strongest. Observed wind directions are almost constant in the box at about 225°. There is a change of the direction between 16:00 and 18:00 UTC for both the model and the observations, which shows the establishment of the NLLJ.

On the one hand, these comparisons with aircraft measurements reinforce our confidence in the model to reproduce adequately the wind speed and direction, thus the main characteristics of pollution transport between Lomé and Savè. On the other hand, this analysis confirms that the PBL depth is not accurately modeled especially in the morning, which could in turn impact the pollution mixing and dilution. During the day, the surface concentration could be over-estimated and the concentration at the PBL top height under-estimated, especially going further away from the sources.

## 4 Inland pollution transport from coast

Firstly, this section investigates the trace gas concentrations at the Savè super-site (Section 4.1). Secondly, we analyze the contribution of the major cities along the coastline to the pollution budget in Savè (Section 4.2). Thirdly, we study how URB plumes and the BB layer observed on 5 July interact at the coast and are transported inland.

### 4.1 Surface pollutant concentrations at Savè

We analyze the temporal evolution of CO, $NO_2$ and NO concentrations. Trace gas concentrations were measured at the ground level at the Savè DACCIWA super-site. We study the hourly temporal variability of observed concentrations over the studied period (1 to 7 July).

During this period, the hourly CO concentration varies between 140 to 250 ppb (Figure 5-top). Moreover, looking at the entire campaign period (25 June to 15 July), hourly CO concentration ranges from 110 to 250 ppb (Figure 14-top). Comparing these two periods, we note that our studied period seems representative for the diurnal cycle over the campaign period. There





is a clear diurnal cycle with the maximum occurring every day at the beginning of the night (between 220 and 250 ppb) and the minimum at the beginning of the day.

Hourly $NO_2$ concentration ranges from 0.2 to 3.5 ppb. We note also that every day there are periods of high $NO_2$ concentrations in the evening and low $NO_2$ concentrations from the morning to the afternoon. It is worth noting that high CO values

noticed in the evening are associated with high $NO_2$ but not with high NO concentrations.

Given the short lifetime of NO (less than one hour) in the PBL (Monks et al., 2009), the analysis of NO concentration gives some clues to understand $NO_2$ variability because NO is mostly linked to local sources (*i.e.* not transported). The baseline of NO concentration is 0.09 ppb (median). High NO concentrations (> 0.5 ppb) are measured on 1 and 7 July in the evening. Moreover, there is an increase every evening, which shows that there are local sources close to the instrument location, probably

associated with charcoal stove cooking or traffic time.

In order to identify periods of high $NO_2$ associated with low NO concentrations, we have computed the $NO_2/NO$ ratio (Figure 5-bottom). This ratio is expected to increase at night by the ozone titration ($O_3$ + NO reaction). During daytime, it depicts local or transported pollution (respectively a low or high $NO_2/NO$ ratio). We note every day a sharp increase in the evening ($NO_2/NO$ > 15), which suggests transported pollutants.

In order to identify periods of high CO associated with low $NO_x$ concentrations, we have computed the $CO/NO_x$ ratio (Figure 5-bottom). When a BB layer reaches the Guinean coast, gaseous nitrogen oxide concentrations are lower than 0.1 ppb (Capes et al., 2009; Reeves et al., 2010), because gaseous nitrogen oxides have been converted into the particulate phase during the transport over the Southeast Atlantic. We therefore expect an increase of CO and constant $NO_x$, when the BB layer reaches Savè without being mixed with URB (containing $NO_x$ in the gaseous phase). At Savè, the $CO/NO_x$ ratio is not higher after

the arrival of the BB layer on 5 July (*cf.* Section 3.1). This result suggests that either the BB layer does not reach Savè in the surface layer or that the BB layer is mostly transported above the PBL.

In order to determine the diurnal cycle of the three pollutants, we present observed hourly averages over 1–7 July together with the maximum and minimum of each hour measured (Figure 6).

There is a clear diurnal cycle of hourly CO concentration averages with the minimum occurring at 08:00 UTC (about 160

ppb) and with the maximum occurring between 18:00 and 22:00 UTC (greater than 200 ppb) over the period 1–7 July (Figure 6) and also over the entire campaign period (Figure 15). This time is in agreement with Adler et al. (2017) and Deetz et al. (2018) who have found using the super-site instrumentation that the coastal front starts moving northward after 16:00 UTC, reaching Savè in the evening. It also corresponds to the highest hourly minimum (190 ppb) and maximum (250 ppb). It is worth noting that CO concentration remains greater than 180 ppb from 22:00 to 04:00 UTC.

For $NO_2$, we also note a clear diurnal cycle with low hourly concentration averages between 08:00 and 15:00 UTC, and with high hourly concentration averages (greater than 1 ppb) from 18:00 to 21:00 UTC over the studied period and also over the entire campaign period (Figure 15). The NO peak at 19:00 UTC is consistent with the usual time of local activities such as traffic and charcoal cook stoves. At 21:00 UTC, there is a high $NO_2$ concentration, as well as a high CO concentration, which is not associated with a high NO concentration, suggesting pollution transport because it is the time of the coastal front passage

(Adler et al., 2017; Deetz et al., 2018).



The NO$_2$ diurnal cycle is similar to the one of CO with a minimum at 08:00 UTC (about 0.6 ppb) and a maximum between 18:00 and 21:00 UTC. The main difference of the NO$_2$ and CO diurnal cycles occurs at night between 21:00 and 02:00 UTC because CO remains high ($\approx$ 200 ppb), whereas NO$_2$ decreases from 1.3 to 0.7 ppb. This result could be linked to a higher ratio of BB compared to URB.

In conclusion, at Savè, there are similar diurnal cycles of CO and NO$_2$ with maxima between 18:00 and 21:00 UTC. Moreover NO concentration is very low at 21:00 UTC, indicating pollution transport from the coastal urban agglomerations and not local production. The BB layer could interact with the URB plumes in the PBL, thus increasing the CO concentration. We need to understand how the BB layer is mixed with URB at the coast, and how it is transported further inland.

### 4.2   Contribution of major coastal cities to the pollution budget at Savè

This section aims at identifying which major cities have a significant contribution to inland pollution at Savè. For this, we analyze the Tracer Experiment 1 described in the models section (*cf.* Table 3).

    In order to present this experiment, the synoptic wind patterns and the pollution plumes of the coastal cities with the URB tracers are displayed on a single figure. The figure represents an average of the modeled plumes over the period 1–7 July in the monsoon layer (from surface to 1 km). Results are presented in an arbitrary unit with the same isocontour value (Iso1) of
tracer concentration for each city (color shadings in Figure 7).

    Over the Gulf of Guinea, we can see a stronger northwestward component of the wind with markedly higher wind speed than over the continent. This figure shows that the pollution plumes of Accra, Lomé and Cotonou could reach Savè, while the direction of the Lagos and Abidjan plumes is not oriented towards Savè.

    We now focus on the temporal variability reproduced by the tracer experiment at Savè (light green dot in Figure 7). The
tracer concentration of the five cities has been interpolated to Savè coordinates (Figure 8-top). The first tracer plume that reaches Savè typically around 19:00 UTC is from Cotonou. In the morning, the Lomé pollution plume reaches Savè, while the Accra pollution plume reaches Savè in the afternoon. There is a short period when hourly concentrations are at a maximum every day, and this peak is associated with the arrival of the Cotonou plume in the evening. This pattern is seen repeatedly over the entire 1–7 July period. The model clearly predicts identified periods when Savè is under the influence of different cities,
which implies that these periods correspond to pollution plumes characterized by different chemical ages.

    From 5 to 6 July, the contribution of Cotonou decreases and the Accra and Lomé contributions increase, which suggests a modification of wind patterns. It is worthy of note that Lagos tracers do not reach Savè, although emissions from Lagos are greater than from the other cities. From midnight to the end of the night, there is no city plume reaching Savè. However, we have seen in Section 4.1 that a high CO concentration persists during the night.

The average diurnal cycle of tracers is presented with the contribution of each city (Figure 8-bottom). It confirms that there are distinct periods when Savè is under the successive influences of Lomé in the morning (06:00 to 12:00 UTC), of Accra in the afternoon (12:00 to 18:00 UTC), and of Cotonou in the evening (18:00 to 01:00 UTC).





These results suggest that the Cotonou plume is affecting Savè during a short period with a maximum between 21:00 and 22:00 UTC (about 2 times greater than the peak magnitude due to Lomé). This is in agreement with observations of CO and $NO_x$ concentrations (*cf.* Section 4.1). We now need to investigate the diurnal cycle of pollutant transport from coastal cities.

### 4.3 Mixing and transport of urban anthropogenic and biomass burning

In this section, results of Tracer Experiment 2 (*cf.* Table 3) are discussed. We present the spatial patterns of URB and BB tracer concentrations averaged over the three layers described in Section 3.2 and then we analyze the vertical structure of these two types of pollution. We focus on 5 July when the BB layer reaches the Guinea Coast (*cf.* Section3.1). The first layer height is 300 m, which is roughly the minimum PBL top height at night.

In order to analyze the interactions between URB emissions and the BB layer, we reproduce the BB layer releasing passive gaseous tracers from 5 July at 00:00 UTC to 6 July at 23:00 UTC with a spatial horizontal extent from 1°W to 2°E at 4.5°N, and the altitude as suggested by satellite observations (Figure 12) at ≈ 1.5 km amsl (*cf.* Section 4.2). Although a non-negligible part of BB is transported in the marine PBL, measurements performed during the DACCIWA campaign have confirmed that the BB layer altitude over the ocean is mostly between 1 and 3 km amsl (Haslett et al., in preparation). URB tracers are not separated by city in this experiment. The threshold of URB tracer concentration for the isocontour presented on the maps is Iso2 = Iso1 × 5 (Figure 9).

On 5 July at 13:00 UTC when shallow dry convection is well developed and looking at the URB in the surface layer, the Accra and Abidjan plumes are transported towards the north, whereas the Lomé, Cotonou and Lagos plumes have a strong eastward component (Figure 9-top-left). This matches the difference in wind direction along the coastline. From 1 to 2 km amsl, the URB tracer distribution is almost the same (with lower wind speed), which shows that the PBL is reaching 1 km amsl at 13:00 UTC. The BB layer emitted at 4.5°N is transported northward between 1 and 2 km amsl over the Gulf of Guinea. BB tracers are not mixed with the marine PBL but with URB tracers from the coast to ≈ 7°N.

On 5 July at 21:00 UTC, when the dry convection has stopped, the model predicts consistent wind speed between the ocean and the continent with a stronger northward component than at 13:00 UTC, especially over the continent. The shape of URB plumes in the first two layers presents two distinct parts, which seems to follow the change of the wind patterns. The area where BB and URB tracers are mixed extends from the coast to ≈ 8°N.

The vertical structure of the wind is now analyzed using cross-sections along a meridional transect from 1°E to 3°E (the red square in Figure 9). We present the same hours of the simulation (13:00 UTC and 21:00 UTC on 5 July), with isocontours Iso2, and other isocontours (Iso3) for both tracers in order to see the core of the plumes: Iso3 = Iso2 × 5.

The vertical structure changes markedly from 13:00 and 21:00 UTC (Figure 10). At 13:00 UTC, shallow dry convection occurs between the coast and ≈ 7°N, which leads to a vertical mixing of BB and URB tracers (Figure 10). The rising motion at the coast is linked to the coastal front that occurs during the day (Adler et al., 2017; Deetz et al., 2018). The sinking motion over the ocean is linked to the land-sea breeze circulation (Knippertz et al., 2017). BB tracers are transported over the marine PBL without mixing. They reach the surface between the coast and ≈ 7°N. URB and BB tracers accumulate along the coastline until the coastal front begins moving northward, which is in agreement with Adler et al. (2017) and Deetz et al. (2018).





At 21:00 UTC, the NLLJ is established at about 400 m amsl from the coast up to 9°N where the front is located. At the coast itself, the front is not present, thus URB and BB tracers are not mixed anymore. The mixture of URB and BB occurring during the daytime is transported northward up to 8°N. The Iso3 isocontour of BB does not reach the surface. It shows that at night the BB layer penetrates further inland. BB and URB plumes are mostly transported above (between 0.5 and 2.5 km amsl) and

within the PBL, respectively.

The discussion in this section is supported by two video supplements (*Click on the links below*):

**1) three layer maps** https://drive.google.com/open?id=1u5DOyUoaKaimcgoqbbfQl8q5OTtGtujy

**2) vertical-meridional transects** https://drive.google.com/open?id=1bcwFYld1KS2-b3AgQBdgzh8xo34nC9Ol

The two videos further illustrate the analysis made at 13:00 and 21:00 UTC on 5 July. They also provide useful additional information to analyze the day to night transition leading to the evening maximum at Savè (Section 4.1). Our simulation reproduces the main features of the diurnal cycle, which are that vertical mixing occurs during daytime, while meridional

advection of pollutants is most efficient at night (Parker et al., 2005).

The coastal front is present from 11:00 to 16:00 UTC from the coast up to ≈ 7°N. At the same time, we notice that URB and BB tracers accumulate along the coastline in the PBL. When dry convection stops, wind speed quickly increases with a stronger northward component. BB and URB tracers are simultaneously transported northward from the coast. From 16:00 to 22:00 UTC, the front is moving towards the north, and the mixture of both BB and URB tracers is advected accordingly. A

similar diurnal evolution of BB and URB transport is simulated on both 5 and 6 July.

The timing of the coastal front propagation in our simulation is in agreement with Adler et al. (2017) and Deetz et al. (2018), who have shown the same regular occurrence of a coastal front that develops during daytime and propagates inland in the evening. After the frontal passage, there is the establishment of the NLLJ (with a jet axis around 250 m amsl), which is also reproduced in our simulation (with an over-estimation of jet axis altitude of about 150 m).

## 25 5 Conclusions

In this study, several observational datasets together with high-resolution model simulation are used to analyze the diurnal cycle of atmospheric pollution transport over SWA. We focus on two distinct pollution sources, urban anthropogenic and biomass burning pollutants (URB and BB respectively), in order to understand their mixing and their advection inland.

We first studied the dynamics and thermodynamics in the lower troposphere over SWA using aircraft, radio-sounding and

ground-based measurements made during part of the DACCIWA field campaign (from 1 to 7 July 2016). In the lowermost troposphere (from the surface to 4 km amsl), the vertical structure of the wind is composed of three layers: (i) from the surface to the PBL top, the wind transports pollutants towards the northeast with an average speed of $\approx 5$ m.s$^{-1}$; (ii) from the PBL top height to 2 km amsl, there is a directional shear layer with low wind speed ($< 4$ m.s$^{-1}$); (iii) from 2 to 4 km amsl, there is the



AEJ layer with high speed of $\approx 8$ m.s$^{-1}$. The three layers are in good agreement with the WRF meteorological simulations at 2-km resolution. Nevertheless, there is a positive bias of wind speed from the surface to 2 km amsl and the modeled PBL height is generally over-estimated, which could affect the dilution of urban plumes, but consequences on pollution transport should be limited to the relative intensity of the different plumes.

The second part of the study uses high-resolution numerical tracer experiments. We analyzed pollution transport from the main urban emission centres of the Guinean coast (Abidjan, Accra, Lomé, Cotonou and Lagos). Observations at Savè show that there is a clear diurnal cycle of NO$_2$ and CO, with a maximum occurring every day between 18:00 and 22:00 UTC, suggesting URB transport from remote emission sites. From the tracer experiments, we demonstrated that there are clear and successive periods of the day when air quality in Savè is affected by different city plumes. Precisely, the contribution of tracers released

from Lomé is greater than 50 % (of the total amount of tracers) between 02:00 to 12:00 UTC, while from 12:00 to 18:00 UTC tracers released from Accra constitute the main contributor. Then, during 3 hours (from 20:00 to 22:00 UTC), tracers released from Cotonou reach Savè leading to a contribution greater than 80 %, while it is lower than 10 % during the other 15 hours (from 03:00 to 18:00 UTC). Over the period, tracers released from Cotonou represent a contribution of 40 %, from Lomé of 36 % and from Accra of 23 %. Our results suggest that the successive periods affected by different city plumes are

characterized by different chemical ages.

To assess the impact of BB on the air quality in Savè, we added a BB layer in the tracer experiments based on satellite observations from MODIS ad CALIOP. The experiment suggest that URB and BB (transported from Central Africa) are mixed along the coastline during the day, whereas at night, URB plumes are transported within the shallow PBL (below about 300 m amsl) and the BB layer is mostly transported between the PBL top and 2 km amsl. The mixture of both URB and BB

accumulated over coastal areas is transported northward in the surface layer from 16:00 UTC onward and reaches Savè (185 km to the north) at 21:00 UTC. Previous studies have already highlighted the importance of the diurnal cycle of the wind along the coastline in the lowermost troposphere over the Gulf of Guinea during the monsoon (Parker et al., 2005; Lothon et al., 2008; Schrage and Fink, 2012; Schuster et al., 2013; Adler et al., 2017; Deetz et al., 2018), and suggested an influence on pollution transport. Our results shows that the coastal front is associated with URB and BB accumulation from 11:00 to 16:00

UTC, and its northward moving is associated with the mixture transport of both pollutants, reaching 8°N at 21:00 UTC.

The WRF simulation reproduces a diurnal cycle of the wind over SWA in agreement with Adler et al. (2017) and Deetz et al. (2018). Indeed, the structure of the wind changes from the morning to the evening. When the shallow dry convection over the land is well developed, wind speed in the PBL reaches a minimum. A coastal front develops during the day and when it ceases, wind speed quickly increases with a stronger northward component. At night, most of the coastal URB plumes are transported

within the planetary boundary layer (below about 500 m above sea level), whereas the BB are mostly transported above it. Our results suggest that BB are generally mixed with urban pollutants emitted along the coastline when it impacts inland ground level air quality.

Over SWA, both wind and URB emissions have a diurnal cycle. The strength of numerical tracer experiments is to enable the dichotomy between the variability linked to the meteorology and the emissions by imposing a constant emission (*i.e.* we

do not account for any diurnal cycle). In this article, only the observed variability linked to the meteorology is analyzed,





we demonstrated that they are clear periods of the day when Savè is impacted by pollution plumes from different cities. In future research, integrated analyses should be conducted to characterize both the URB plumes and the BB layer in terms of composition, gaseous and particulate phase, oxidation of the organic components, and spatio-temporal variability. The DACCIWA campaign provides unique and valuable observations that will allow the investigation of the perspectives opened by this article based on tracer experiments.

*Acknowledgements.* The research leading to these results has received funding from the European Union 7th Framework Programme (FP7/2007-2013) under Grant Agreement no. 603502 (EU project DACCIWA: Dynamics-aerosol-chemistry-cloud interactions in West Africa). The Service des Avions Français Instrumentés pour la Recherche en Environnement (SAFIRE, a joint entity of CNRS, Météo-France and CNES and operator of the ATR 42), the British Antarctic Survey (BAS, operator of the Twin Otter) and the Deutsches Zentrum für Luft- und Raumfahrt (operator of the Falcon 20) are thanked for their support. We acknowledge Claire Delon and Fabienne Lohou (Laboratoire d'Aérologie) for helpful discussions.





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

**Appendix A: Supplemental Material**




## List of Figures









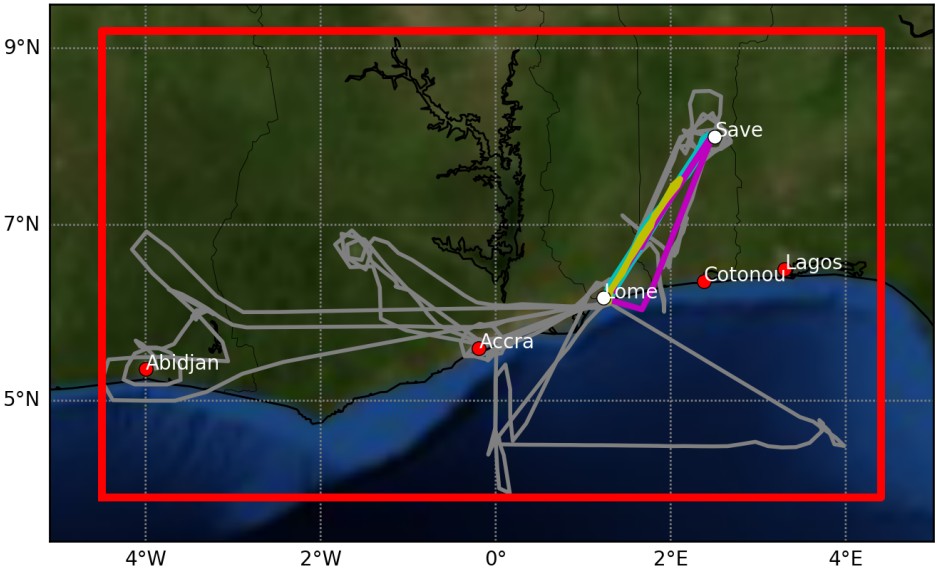

**Figure 1.** *Map of the modeling domain (red rectangles) with location of the major cities (red dots), of the Lomé airport and the Savè super-site (white dots). Superimposed are the flight tracks of the three research aircraft during the 1-7 July 2016 period (grey lines). The aircraft flight tracks on 5 July are colored for the German Falcon (blue line), the French ATR-42 (violet line) and the British Twin Otter (yellow line).*



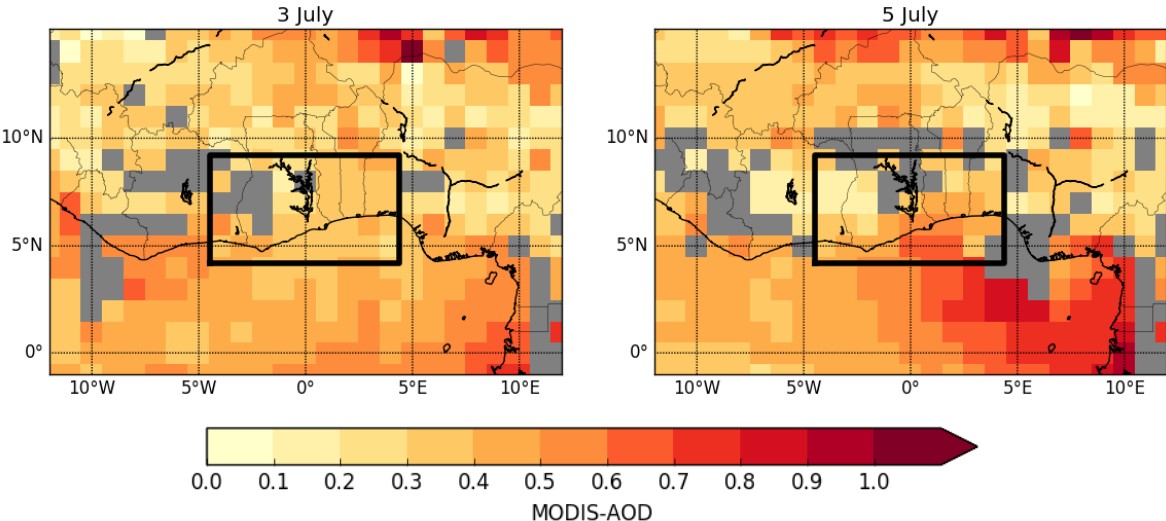

**Figure 2.** *MODIS-AOD 1-day moving average of two products acquired by Aqua and Terra (the combined Dark-Target and Deep-Blue MYD08-D3 and MOD08-D3 products respectively) on 3 July 2016 (left) and 5 July 2016 (right). Data excluded by the cloud screening process are in gray. The modeling domain is presented by the black square.*





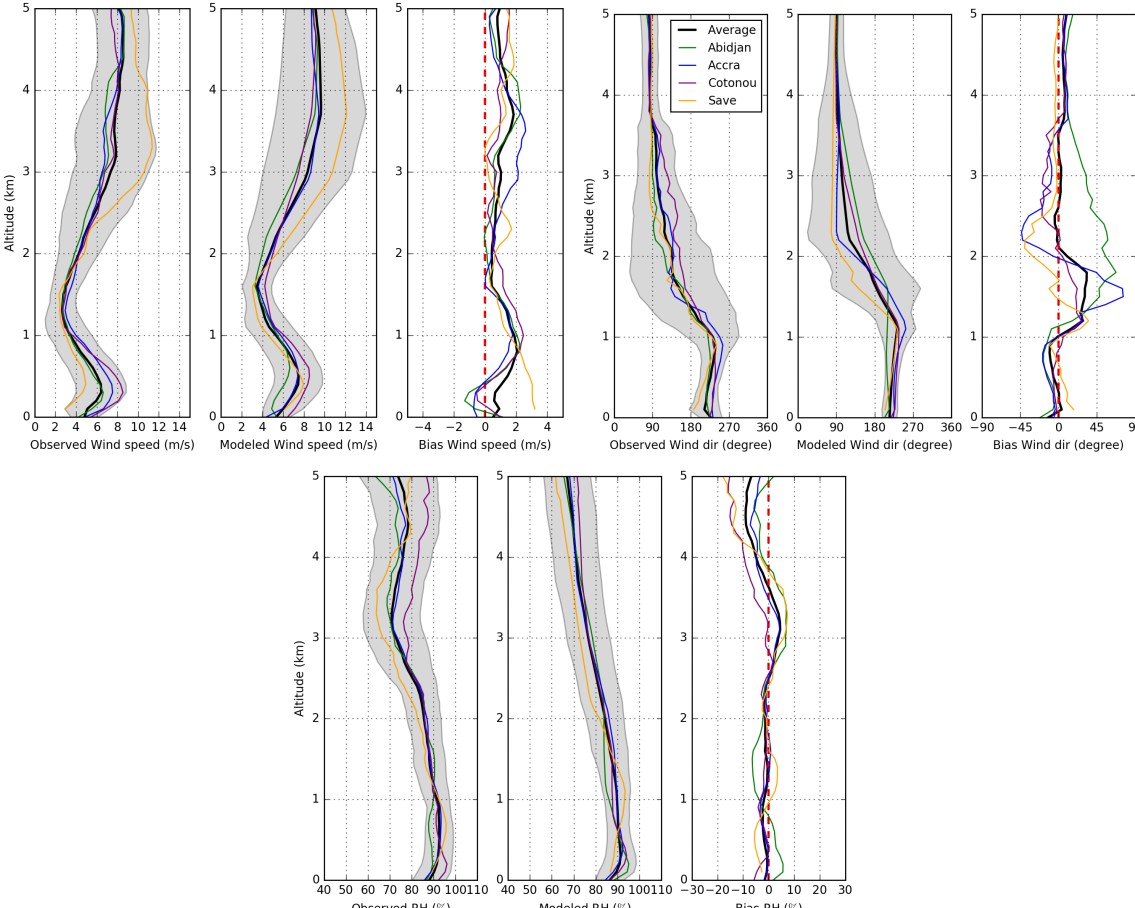

**Figure 3.** *Observed and modeled mean vertical profiles of wind speed (in m.s$^{-1}$) and direction (360° circle with 0° and 360° is the north and 90° is the east), and relative humidity (RH in %) averaged of all profiles over the period 1-7 July 2016 at Abidjan in Ivory Coast (green line), Accra in Ghana (blue line), Cotonou in Benin (purple line) and Savè in Benin (orange line). The mean and standard deviation at the four locations are represented by the black line and the gray shading. The right panel presents the (mod-obs) mean vertical bias of each location and of the average of the four locations.*





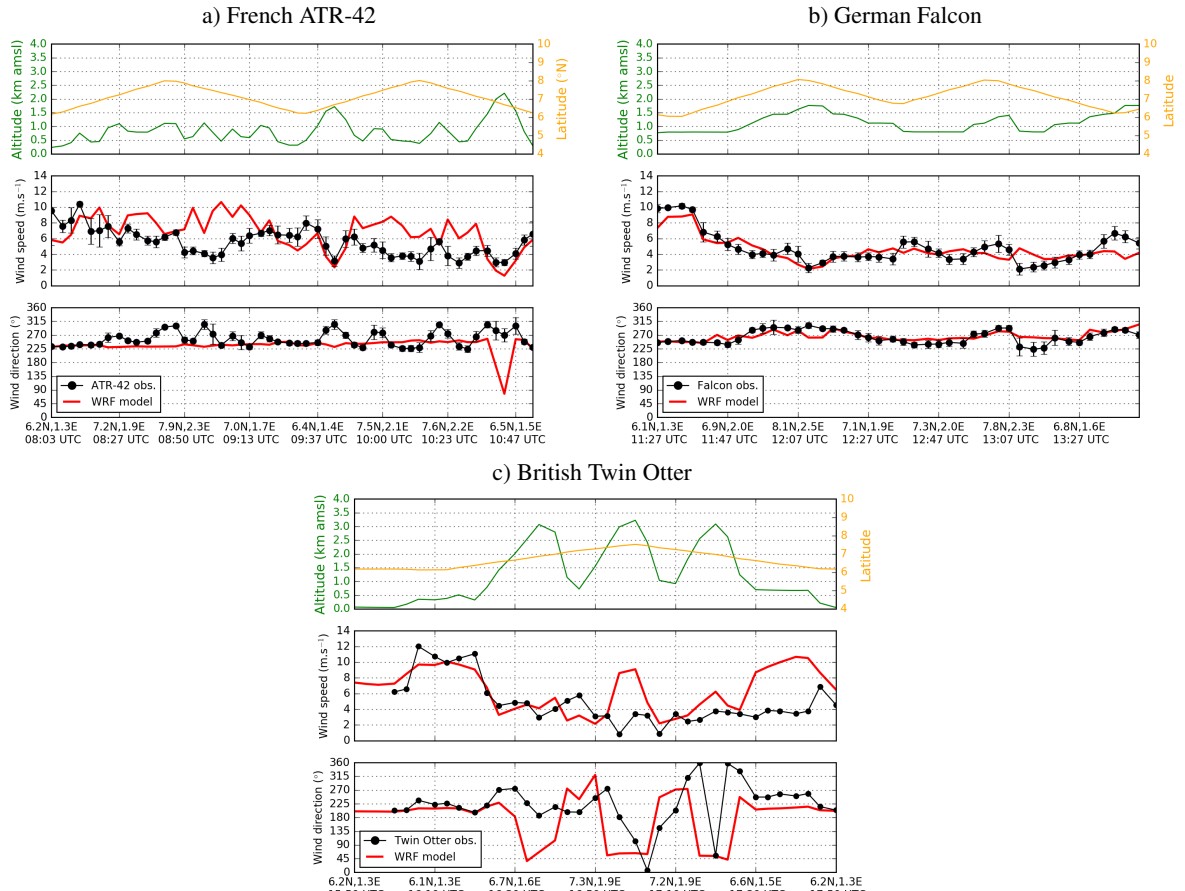

**Figure 4.** *Time series on 5 July 2016 of (a) the French ATR-42, (b) German Falcon and (c) British Twin Otter aircraft data, composed of three panels for each aircraft: (top) altitude (in m) with the latitude (in °N), (middle) wind speed (in m.s$^{-1}$) and (bottom) direction (in degree). Modeled values with the WRF model are interpolated along the flight positions (red line). Observed value averages are the black dots with the hourly standard deviation (error bars).*





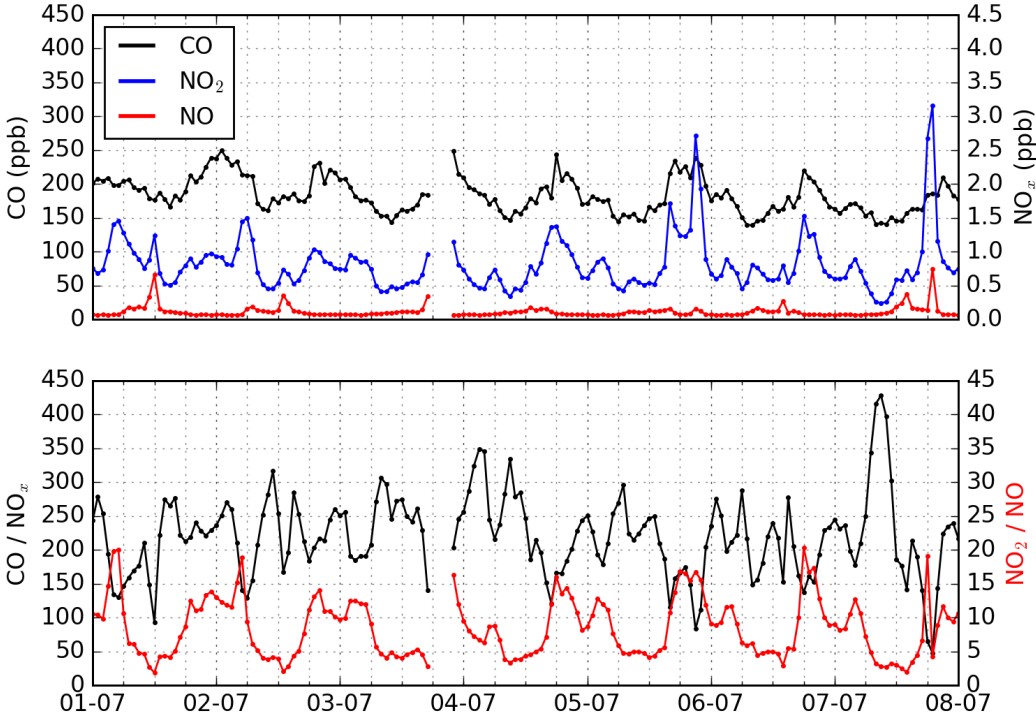

**Figure 5.** *Time series of (top) carbon monoxide (CO), nitrogen dioxide (NO₂) and nitrogen monoxide (NO) hourly concentration averages (in ppb) observed at Savè (Benin) for the period 11–7 July 2016, and (bottom) of the ratios: CO/NOₓ (in black) and NO/NO₂ (in red).*





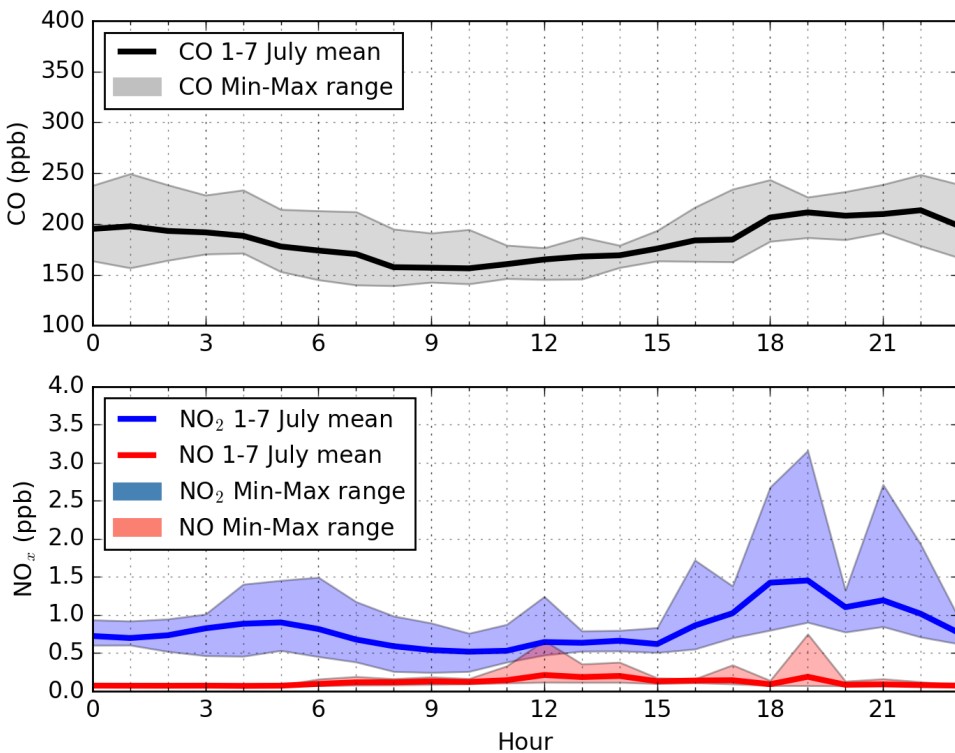

**Figure 6.** *Hourly diurnal cycles of (top) carbon monoxide (CO in black) and (bottom) nitrogen dioxide (NO₂ in blue) and nitrogen monoxide (NO in red) concentrations (in ppb) observed at Savè (Benin). Means of each hour are presented by the lines over the period 11–7 July 2016 and the upper and lower shading limits correspond to the hourly ranges (maximum and minimum of each hour over the period).*




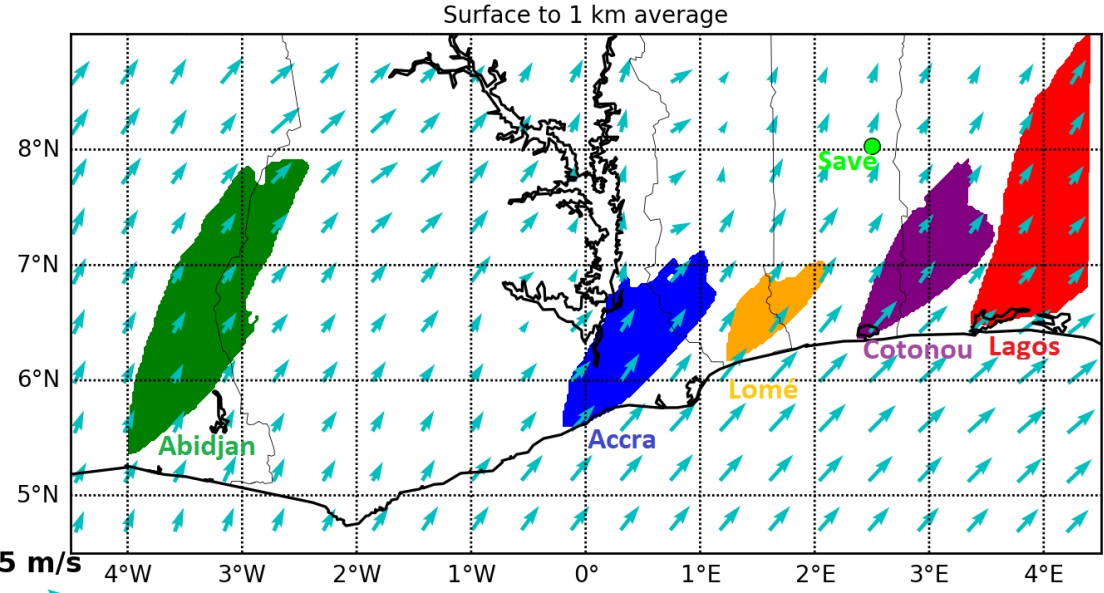

**Figure 7.** *Maps of URB tracer concentration (in arbitrary unit) averaged over the period 1 to 7 July 2016 in the monsoon layer (from the surface to 1 km amsl). Tracers are released from Abidjan (Ivory Coast) in green, from Accra (Ghana) in blue, from Lomé (Togo) in orange, from Cotonou (Benin) in violet, from Lagos (Nigeria) in red. The same threshold of tracer concentration is used for all city plumes for the color shading (Iso1). Wind vectors at 10 m modeled by WRF are presented by the light blue arrows. The location of the DACCIWA super-site (Savè in Benin) is presented by the light green dot.*





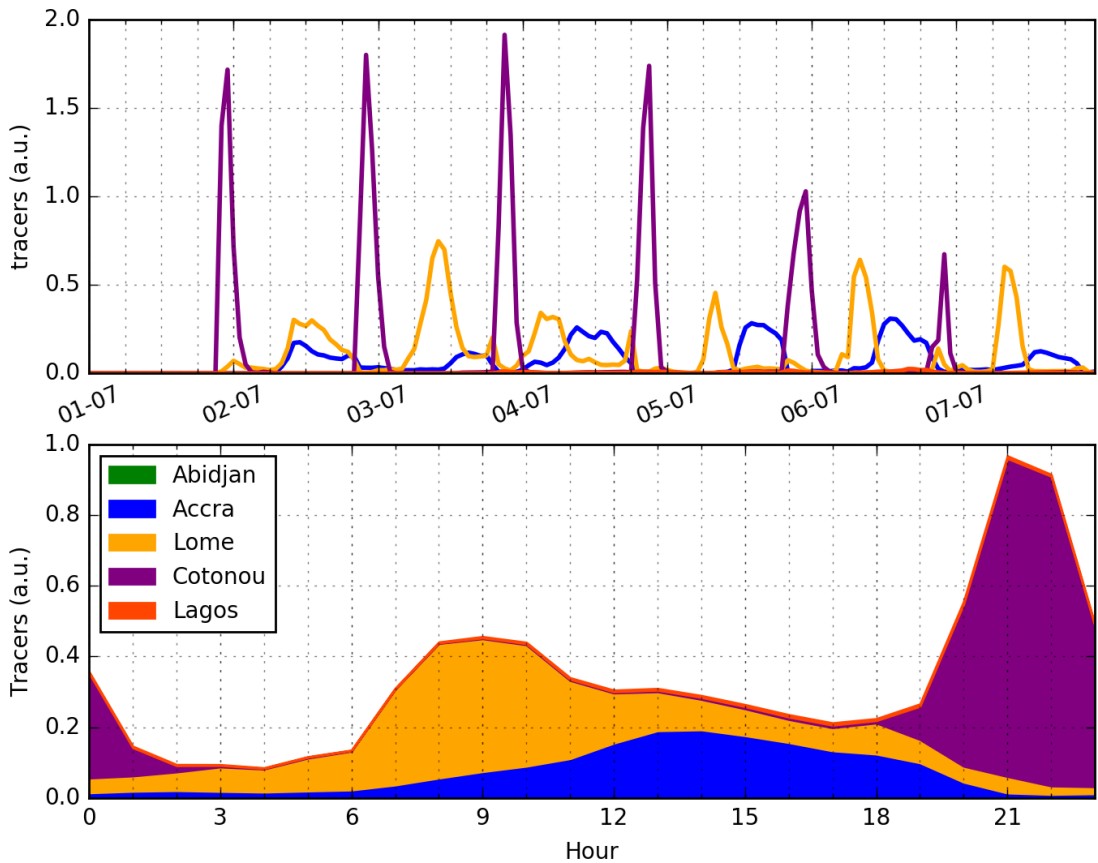

**Figure 8.** *Time series of hourly URB tracer concentrations (in arbitrary unit) modeled by the CHIMERE model at the DACCIWA field campaign ground station in Savè (Benin) for the period 1-7 July 2016. Urban tracers are released from five coastal cities: Abidjan (Ivory Coast) in green, Accra (Ghana) in blue, Lome (Togo) in orange, Cotonou (Benin) in violet, and Lagos (Nigeria) in red.*

©c Author(s) 2018. CC BY 4.0 License.







**Figure 9.** *Maps of concentrations of URB (in gray) and BB (in brown) numerical tracers on 5 July 2016 at 13:00 UTC (left) and at 21:00 UTC (right). Concentrations are averaged over three layers (top) from the surface to 300 m amsl, (middle) from 0.3 to 2 km amsl, (bottom) from 2 to 4 km amsl. Wind vectors at 10 m from the WRF model are presented by the light blue arrows. The meridional-vertical transect shown in Figure 10 corresponds to the zonally averaged area of the red rectangle.*





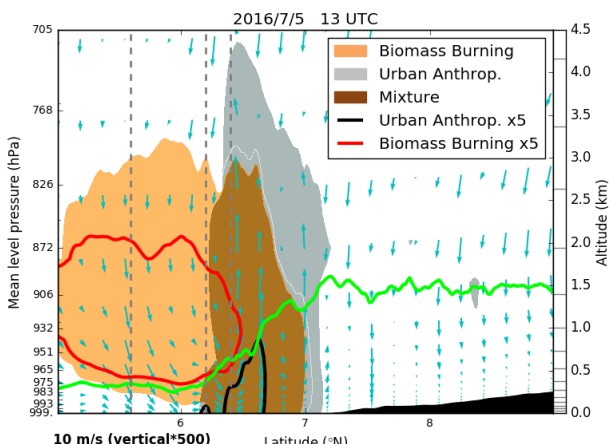
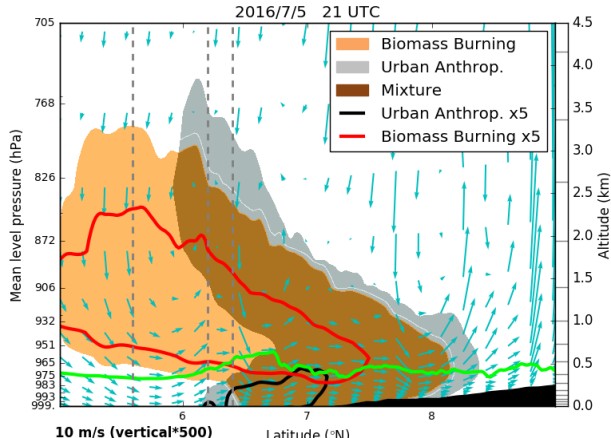

**Figure 10.** *Meridional-vertical transect of concentrations of URB (in gray) and BB (in brown) numerical tracers on 5 July 2016 at 13:00 UTC (left) and at 21:00 UTC (right). PBL height from the WRF model is the green line. The vertical dashed gray lines show the latitude of Accra (5.6° N), Lomé (6.2° N) and Cotonou (6.4° N). The black area is the topography. Vectors (light blue arrows) represent wind in the plan of the transect (with an aspect ratio of 500 between the meridional and vertical lengths).*

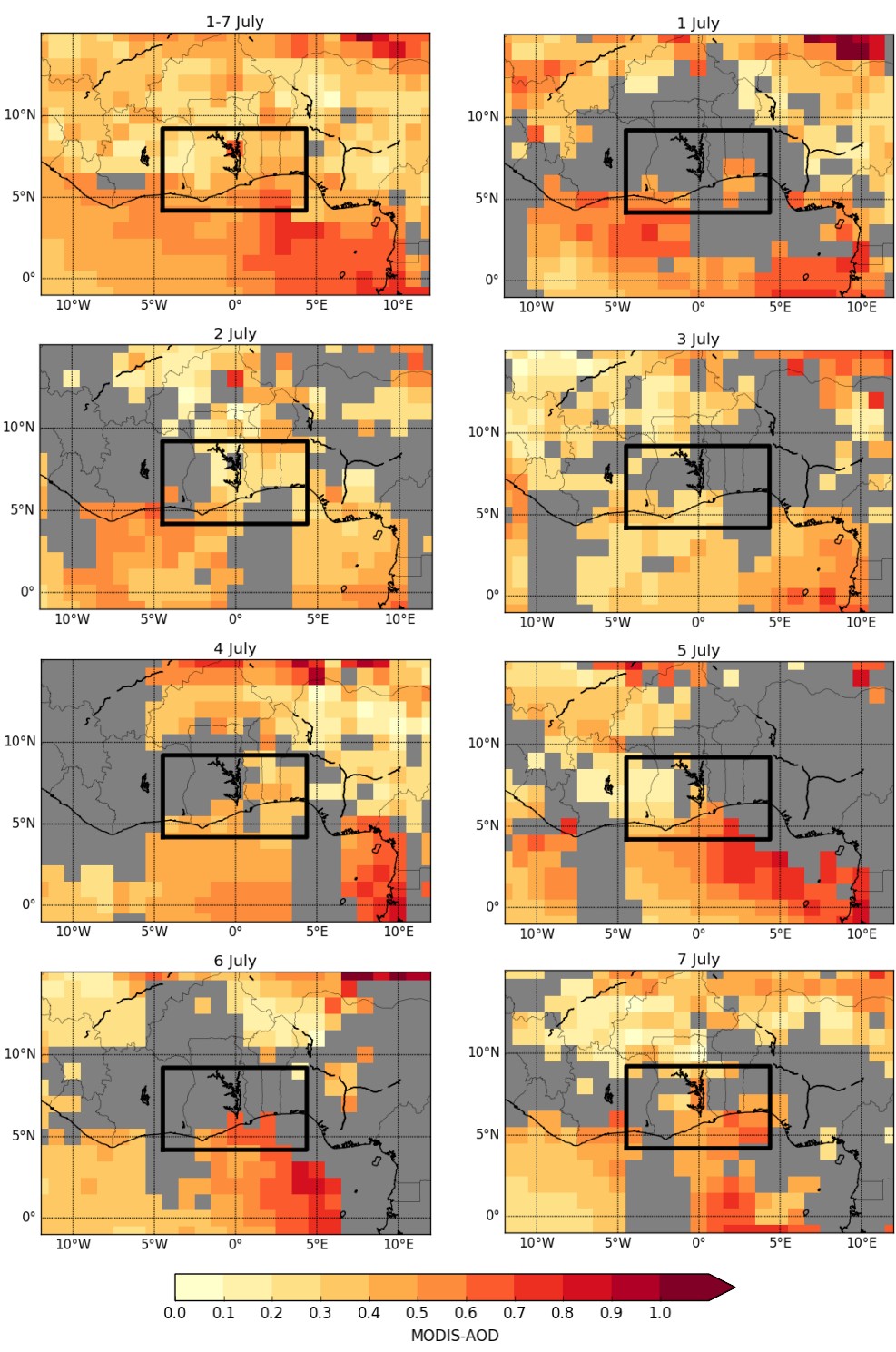

**Figure 11.** *MODIS-AOD daily average of two products acquired by Aqua and Terra (the combined Dark-Target and Deep-Blue MYD08-D3 and MOD08-D3 products respectively) over the 1-7 July 2016 period (top-left) and over each day of the period. Data excluded by the cloud screening process are in gray. The modeling domain is presented by the black square.*



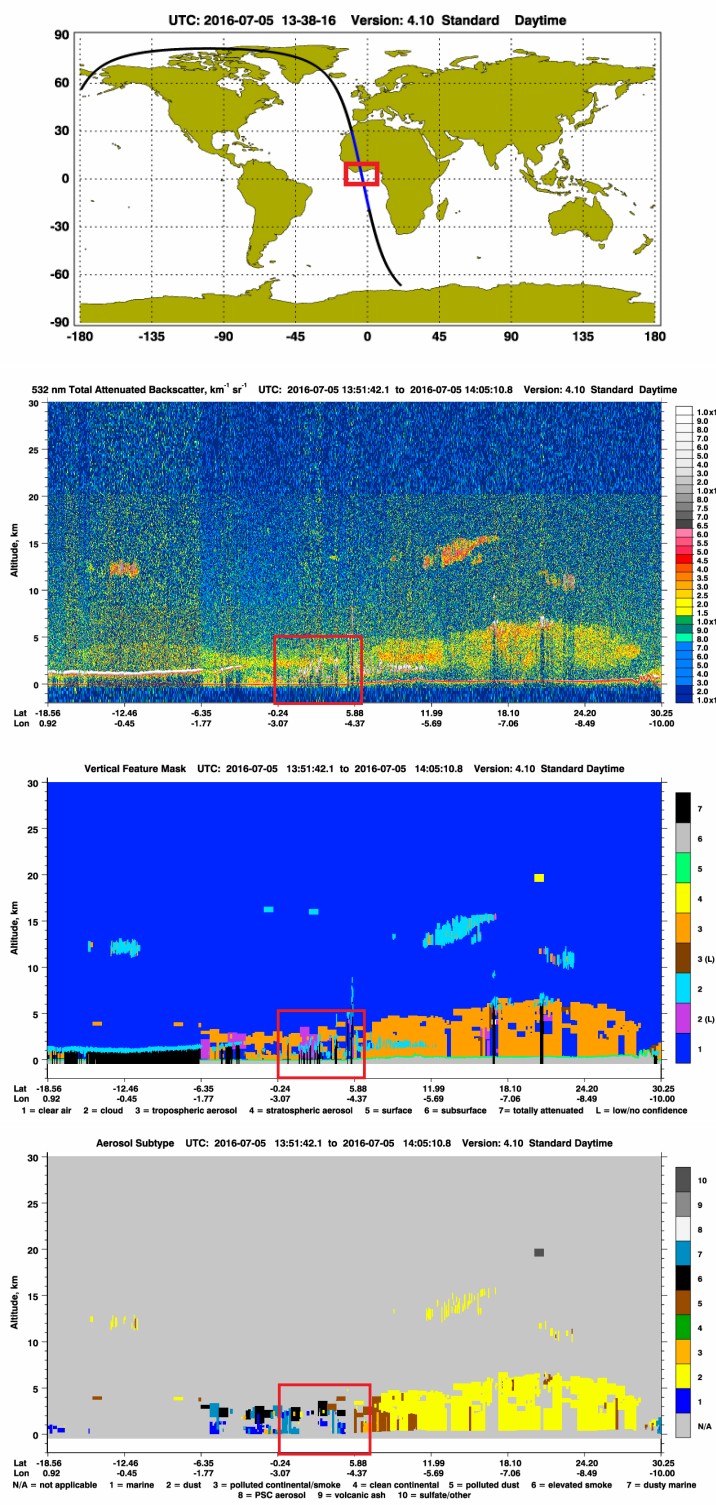

**Figure 12.** *Space Lidar CALIPSO data onboard the CALIOP platform on 5 July 2016 with the orbit track location, the vertical profile of the total attenuated backscatter, the vertical feature mask and the aerosol subtype. The modeling domain is the red box. (data are available on www-calipso.larc.nasa.gov/products/lidar/browse_images).*





**Figure 13.** *Wind speed (top-left panel) and direction (top-right panel) hourly averages and standard deviations on 5 July 2016 for all aircraft measurements (dots and errorbars) acquired in a box (latitude from 6.6° N to 7.8° N, in longitudes from 1.5° E to 2.2° E, in altitude from 300 m to 1000 m). The horizontal extent of the box and the aircraft tracks are displayed on the map (bottom panel). Modeled value averages of the whole box are presented by the red line (and standard deviation by red shading). Blue dots correspond to ATR-42 measurements, violet to Falcon and yellow to Twin Otter.*





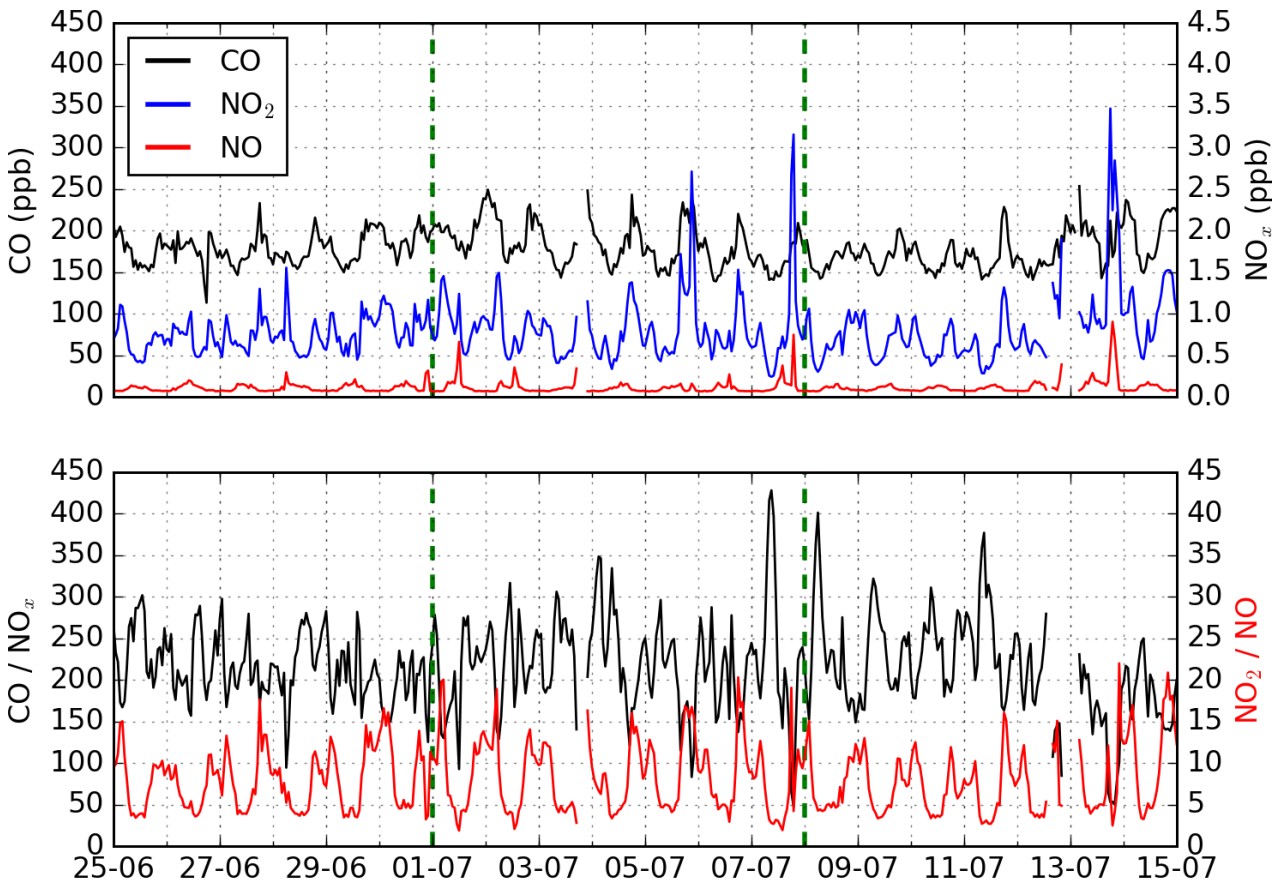

**Figure 14.** *Time series of (top) carbon monoxide (CO), nitrogen dioxide (NO$_2$) and nitrogen monoxide (NO) concentrations (in ppb) observed at Savè (Benin) over the entire campaign period (25 June to 15 July 2016), and (bottom) of the ratios: CO/NO$_x$ (in black) and NO$_2$/NO (in red). The studied period (1-7 July 2016) is defined by the two green dashed vertical lines.*




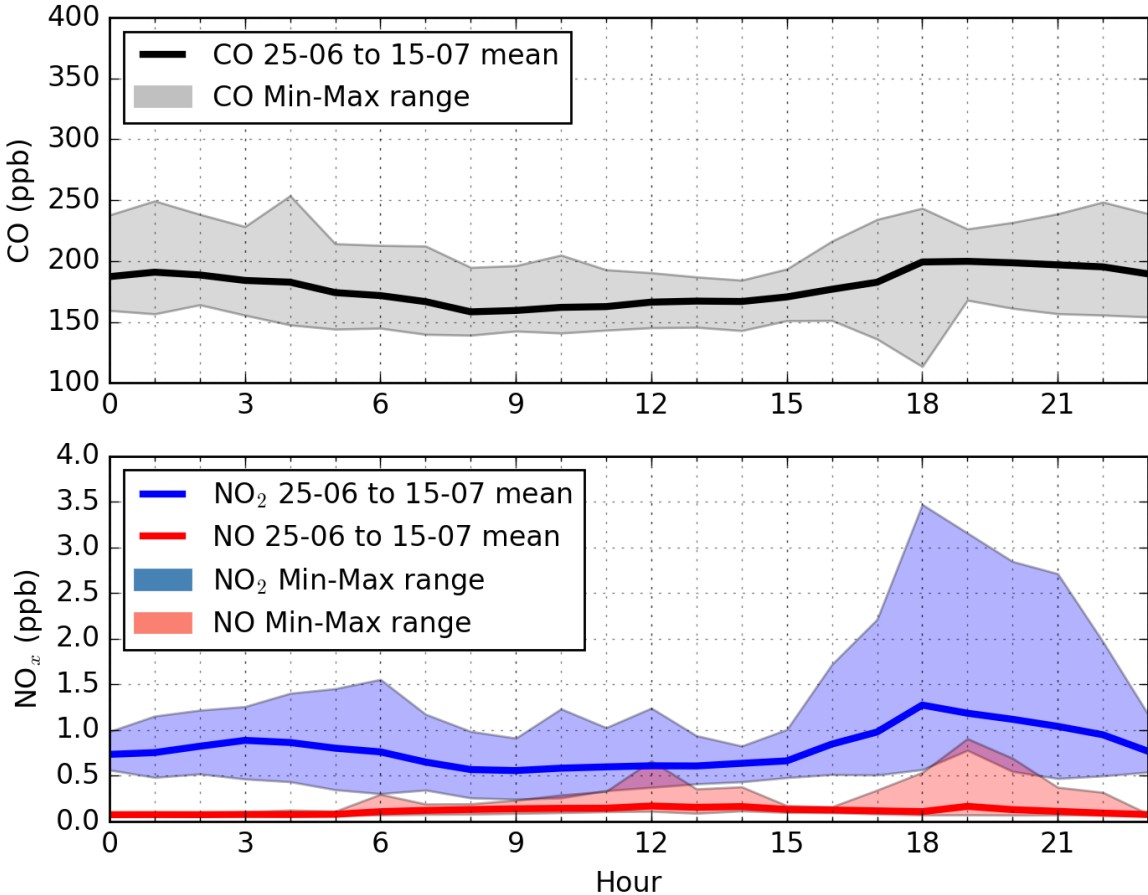

**Figure 15.** *Hourly diurnal cycles of (top) carbon monoxide (CO in black) and (bottom) nitrogen dioxide (NO$_2$ in blue) and nitrogen monoxide (NO in red) concentrations (in ppb) observed at Savè (Benin). Means of each hour are presented by the lines over the entire campaign period (25 June to 15 July 2016). The upper and lower shading limits correspond to the hourly ranges (maximum and minimum of each hour over the period).*



## List of Tables



| City | Country | Latitude | Longitude | Elevation | Number of inhabitants |
|-------|------------|---------|----------|-------------|------------------------|
| Abidjan | Ivory Coast | 5.36°N | 4.00°W | 50 m amsl | **4 923 000** |
| Accra | Ghana | 5.60°N | 0.19°W | 30 m amsl | **3 013 000** |
| Lomé | Togo | 6.17°N | 1.23°E | 10 m amsl | **1 830 000** |
| Cotonou | Benin | 6.36°N | 2.38°E | 10 m amsl | 2 194 000 |
| Savè | Benin | 8.03°N | 2.49°E | 130 m amsl | 87 000 |
| Lagos | Nigeria | 6.49°N | 3.36°E | 10 m amsl | **13 121 000** |

**Table 1.** *Characteristics of the studied cities with: country, latitude, longitude, elevation above mean sea level (amsl), and number of inhabitants of urban agglomerations. The population in bold is given for 2015 according to World Urbanization Prospects (The 2011 Revision). National general population and habitat census is used to estimate the population of Cotonou and Savè (2013 Census).*



| Datasets | Platform | Variables | Frequency |
|---|---|---|---|
| Ground based station | Savè super-site<br>operated by KIT-UPS universities | $NO_2$, NO<br>and CO concentrations | Raw data: 1 hz<br>Presented: hourly averages |
| Aircraft | ATR-42, Twin Otter, Falcon<br>operated by SAFIRE, BAS and DLR teams | Relative humidity<br>Wind direction and speed | Raw data: 1 hz<br>Presented: 3-min averages |
| Radiosonde | Launch sites:<br>Abidjan, Accra, Cotonou, Savè | Wind direction and speed<br>Relative humidity | High resolution 1 hz<br>Presented: 100 m averages |
| Satellite | MODIS on Terra<br>and Aqua | AOD (550 nm)<br>level 3 ($1°x1°$) | Daily |

**Table 2.** *Datasets used in this study with acquisition platform, variables and sampling frequency.*



| | Tracer Experiment 1 | Tracer Experiment 2 |
|---|---|---|
| Tracer type | URB tracers only | BB and URB tracers |
| Release duration | 1-7 July | 5-7 July (BB) and 1-7 July (URB) |
| Release altitude | lowest level | at 1500 m (BB) and lowest level (URB) |
| Release location | 5 cities | from $1°$W to $2°$E at $4.5°$N (BB) and 5 cities (URB) |
| Number of tracer | 5 (each city) | 2 (BB and URB) |

**Table 3.** *Main characteristics of the two numerical tracer experiments using high resolution modeling at 2-km grid spacing with tracer emissions relevant for biomass burning (BB) and urban pollutants (URB).*



| Var | N | Q1 | | Median (Q2) | | Q3 | | Mean | | Bias | |
|---|---|---|---|---|---|---|---|---|---|---|---|
| | | Obs | Mod | Obs | Mod | Obs | Mod | Obs | Mod | Absolute | Relative (%) |
| **0 to 1 km** | | | | | | | | | | | |
| **RH** | 349 | 90.75 | 85.24 | 94.80 | 88.65 | 98.61 | 93.50 | 94.19 | 88.12 | -6.07 | -6 % |
| **W. speed** | 349 | 3.80 | 3.98 | 5.25 | 5.55 | 6.47 | 6.80 | 5.31 | 5.51 | 0.19 | 4 % |
| **W. dir.** | 349 | 220.08 | 214.17 | 229.43 | 228.47 | 244.41 | 237.67 | 231.78 | 228.19 | -3.59 | -2 % |
| **1 to 2 km** | | | | | | | | | | | |
| **RH** | 116 | 91.41 | 88.10 | 95.58 | 90.85 | 98.77 | 94.79 | 94.33 | 90.86 | -3.47 | -4 % |
| **W. speed** | 116 | 2.10 | 1.88 | 2.90 | 2.80 | 4.04 | 3.94 | 3.30 | 3.04 | -0.26 | -8 % |
| **W. dir.** | 116 | 191.17 | 207.29 | 244.71 | 246.59 | 285.29 | 272.28 | 228.56 | 228.41 | -0.15 | >1 % |
| **2 to 4 km** | | | | | | | | | | | |
| **RH** | 62 | 76.34 | 73.99 | 84.43 | 76.86 | 90.44 | 85.18 | 76.12 | 79.63 | 3.51 | 5 % |
| **W. speed** | 62 | 3.34 | 2.23 | 4.71 | 5.69 | 9.08 | 9.75 | 6.37 | 6.67 | 0.30 | 5 % |
| **W. dir.** | 62 | 55.73 | 45.76 | 72.12 | 69.04 | 137.67 | 145.84 | 107.36 | 105.24 | -2.12 | -2 % |

**Table 4.** *Observed and modeled distribution (first quartile, median, third quartile), mean and bias (absolute and relative) of relative humidity (%), wind direction (deg) and speed (m.s$^{-1}$) measured by the three aircraft over the period 11–7 July 2016 separated into three altitude ranges: surface to 1 km, 1 to 2 km and 2 to 4 km amsl, respectively.*