# Peer review of "Diurnal cycle of coastal anthropogenic pollutant transport over southern West Africa during the DACCIWA campaign"

_Atmospheric Chemistry and Physics, 2018_

## Referee Comment (RC1) · Anonymous Referee #1 · 30 Aug 2018

**Deroubaix et al., 2018, ACPD, Diurnal cycle of coastal anthropogenic pollutant transport over southern West Africa during the DACCIWA campaign**

General Description:
The authors use surface and aircraft observations, radiosondes, satellite observations, and a model to determine the diurnal cycles of $NO_x$ and CO at a short-term monitoring site in southern West Africa and assess the contribution of individual sources and synoptic-scale meteorology to this diurnal variability. The figures are clearly presented and the model tracer simulation experiment is informative of the varying contribution of pollution from cities and other sources to pollution at the monitoring site. The content is appropriate for ACP, but there is limited context for the relevance of the results for other time periods and locations in southern West Africa and the implications of findings for future air quality due to increasing urbanisation and changes in biomass burning (Andela et al., 2014). It is for this reason that I'm hesitant to accept as is and highly recommend that the authors strengthen the relevance of the paper by addressing its limited scope.

General Comments:
How does the paper fit within the context of other findings from analysing observations and model output during DACCIWA, and also enhance understanding of the region since the AMMA campaign?

There are many typos that can be eliminated with a careful reread (extents on P2, L5; 21th on P2, L9; acquiered on P4, L18 etc.).

Please fix incorrect order of in-text citations when there is more than one article in the same year by the same first author (P2, L14; P2, L27 etc.).

Why does the study focus on Savè, beyond logistics? Is there rapid population increase? Is it an ideal location to understand synoptic scale meteorological patterns in southern West Africa? This could be better justified.

Only 7 days of observations are considered. Can we draw conclusions about an extended time period based on this brief analysis period? And if so, what time period? The full year? The entire monsoon period? The onset period only?

Absent from the study are measurements and/or a discussion of non-methane volatile organic compounds (NMVOCs) and aerosol mass concentrations and composition. Emissions of these are high from local sources and from distant biomass burning (Liousse et al., 2014; Marais et al., 2016; Janicot et al., 2008; Reeves et al., 2010). Can the same conclusions about sources and diurnal variability be drawn about these? Were these measurements made during DACCIWA on the aircrafts flown? If so, do these offer any utility in understanding these pollutants or confirming similar diurnal behavior? If not, are there other studies that could be referenced to assess the sources of these pollutants and the implications for air quality?

Change MODIS-AOD to MODIS AOD throughout.

The figures are presented out of sequence in the text (e.g. Figures 11 and 12 are mentioned before Figure 3). Please reorder the figures so that these are introduced sequentially.

Include a link to the MODIS AOD product used in this work (preferably doi, otherwise URL).

Shorten the conclusion to the main findings of the paper, rather than providing a detailed overview of the methods, results, and outcomes.

Specific Comments:
P3, L12: Consider instead referencing a relevant publication from AMMA or from analysis of MOZAIC vertical profiles that finds influence of Central African biomass burning on atmospheric composition in southern West Africa.

Table 1: The table title says 2015 population but from a 2011 Revision. Are these projected estimates? Why not use a recent revision? No reference is provided for the 2011 Revision, so it's not possible to assess what the 2015 population is when it was revised in 2011.

Section 2.2: Are the dashed indented lines headers? These are unnecessary. Instead integrate these in the relevant paragraph and make clear in the first sentence what the paragraph is about.

P5, L9: How frequently are the radiosondes launched during the time period considered in this study?

P5, L12-17: How are biomass burning layers identified with a total column AOD measurement?

P5, L20: How accurate is CALIOP at distinguishing aerosol types?

P5, L28-29: Knippertz et al. (2017) isn't the seminal paper on defining the monsoon onset period. Do the authors mean to say that the monsoon onset period specific to DACCIWA (2016) is 22 June to 20 July?

P6, L23-30: Elaborate on the pseudo-anthropogenic emissions that are imposed. Are these based on an existing inventory? How much is emitted per one million inhabitants? Does the imposed magnitude of emissions matter?

P7, L9-11: Was there any rain during 1-7 July that would cause aerosols to wet deposit and so affect interpretation of output from a model that does not include sinks?

P7, L26-28: This transport pattern has already been observed during AMMA and before that using MOZAIC aircraft campaign observations. Consider including these studies too to reinforce that this work builds on previous research and campaigns.

Figure 3: How representative is the site of the coincident 2 km x 2 km model gridsquare?

Section 3.2: "From research aircraft" and "From radiosondes" are not complete headers. Consider revising.

P11, L1: Why define this "feature G". It's not used again and doesn't seem to appear in any of the figures.

P12, L30-35: Does interference in the $NO_2$ measurement from dissociation of $NO_x$ reservoir compounds (Reed et al., 2016) impact the interpretation of $NO_2$ and $NO_2/NO$ diurnal variability?

P13, L27-29: This result isn't surprising, as the monsoon flow is prevailing southwesterly and so transport from Lagos isn't expected. It would be more noteworthy if this did occur.

P14, L1: What is the implication on future air quality that the Cotonou plume affects Savè?

Videos: Consider adding annotations or narration to the videos to guide the viewer. Also consider adding time stamps in the text to point the reader to these specific features in the videos.

References:
Andela et al., doi:10.1038/nclimate2313, 2014.
Janicot et al., doi:10.5194/angeo-26-2569-2008, 2008.
Knippertz et al., doi:10.5194/acp-17-10893-2017, 2017.
Liousse et al., doi:10.1088/1748-9326/9/3/035003, 2014.
Marais et al., doi:10.1021/acs.est.6b02602, 2016.
Reed et al., doi:10.5194/acp-16-4707-2016, 2016.
Reeves et al., doi:10.5194/acp-10-7575-2010, 2010.

---

## Referee Comment (RC2) · Anonymous Referee #2 · 27 Sep 2018

General Description

The authors used ground-based and aircraft measurements of meteorological and chemical (CO, NO, NO2) variables collected during DACCIWA campaign to understand the transport pathway of biomass burning and coastal urban emissions in SWA. To further investigate this, they designed and used two experiments using WRF-CHIMERE (in tracer mode). First, they thoroughly studied the model performance and its limitations in capturing the transport. Next, they used the tracer experiments to assess the contribution of urban and biomass burning pollutions in the region. The manuscript is well written, and I recommend it for ACP after fixing the typos and addressing some

issues listed below.

General Comments: - There are mistakes in the order of references and format of the references in the introduction. Please double check the references.

- UTC and local time description (morning/evening) were used interchangeably. Please provide information on the time zone and sunset/sunrise time.

- The section on population can be shortened as some parts in introduction and 2.1 overlaps.

- How does the result compare with the AMMA campaign?

- The order of the figures is not correct. Please use supplement material if necessary.

- In section 3.2.2. the discussion on model performance in capturing wind and RH is difficult to follow. Please either discuss layer-by-layer (starting from the lowest layer) or variable by variable.

- In section 3.2.2. How does the result compare with radiosondes results?

- Estimating PBL height from radiosonde measurements (separating day and night measurements) and comparing with model PBL height can be beneficial for the discussion in section 3.3

- I suggest adding more details on the PBL height and its influence on the concentration of different constituent, especially during nighttime.

Specific Comments:

P3-L1-5: I suggest adding more references on the source attribution to provide a bigger picture. What is the contribution of these sectors in other regions or bigger domains? For example, compare with Sobhani et al., 2018; Yang et al., 2017; Kulkarni et al., 2015

P3-L2: What year? 2006?

P3-L12: "Haslett et al., in preparation" I don't think this is an acceptable format and it

is not mentioned in the References section.

P3-L18: "1-7 July" Please add 2016.

P4-L9: the Lome population stated in the text does not match Table 1.

P4-L10 and the next paragraph: Information given on population of Cotonou is confusing. Is it necessary to state 1,086,00 inhabitant and then change it in the next paragraph?

P42-Table 2: Please provide coordinates of the ground sites.

P5-L9: What is the local time?

P5-section 2.3: Please add a figure of outer domain.

P6-L2: How many layers below 1 km?

P6-L5: Please reference GFS data.

P6-L5: Did you use GFS data only for nudging or for meteorological initial and boundary conditions as well? Please clarify in the text.

P6-L14: Did you use cumulus parametrization for both domains?

P6-L27: What time do you start releasing the tracers? Have you allowed for spin up in the model before releasing the tracers?

P6-L28: "grid cell of each city". One grid cell? Given the high resolution of your inner domain, did you release tracers from one grid cell or from a region (city)? Please clarify in the text.

P7-L1: How much tracer did you release? Please clarify how you "reproduce the BB layer observed with MODIS" considering that the tracers are gaseous.

P9-L16: "Section 3.2" Do you mean Section 3.2.1?

P9-L18: Please comment on RH in layer (ii) to be consistent.

[Figure]

P9-L24: "the first kilometer" is this the monsoon layer? Please be consistent.

P9-L27: "wind speed is wider than at lower altitudes". Remove "than".

P10-L18: Did you compare measured and modeled RH? This can help to better understand the model performance in capturing PBL height and (maybe) clouds.

P11-L1: What is "feature G"?

P11-L22: NO and NO2 have a short lifetime and are not trace gases.

P29-Figure 5: Do thick marks indicate 00Z?

P12-L14-15: Increase in evening NO2/NO can be due to lower PBL (higher NO2 concentration) and reduction in NO by reacting with O3.

P12-L20: On July 5th concentration of both CO and NOx (more) increased in the evening thus resulted in a decrease in CO/NOx ratio and increase in NO2/NO. Maybe a mixture of BB and URB was transported to Save?

P12-L23: no need to start a new paragraph. P12-28-29: Could the nighttime increase in CO (and NO2 in the next paragraph) concentrations be due to lower PBL height? Please comment in the text.

P12-L32: What about NO peak at 12UTC?

P12-L34: Again, lower nighttime PBL height can also justify the increase in CO and NO2 concentrations.

P14-L4: or shorter lifetime of NO2?

P13-L13: please reference the figure after "single figure"

P13-L16: "northwestward" you mean "southwestward"?

P13-L21: What time did you start releasing the tracers?

References: - Yang, Y., Wang, H., Smith, S. J., Ma, P.-L. and Rasch, P. J.: Source

attribution of black carbon and its direct radiative forcing in China, Atmos. Chem. Phys., 17(6), 4319–4336, doi:10.5194/acp-17-4319-2017, 2017.

-Kulkarni, S., et al. "Source sector and region contributions to BC and PM 2.5 in Central Asia." Atmospheric Chemistry and Physics 15.4 (2015): 1683-1705.

- Sobhani, N., Kulkarni, S. and Carmichael, G. R.: Source Sector and Region Contributions to Black Carbon and PM2.5 in the Arctic, Atmos. Chem. Phys. Discuss., 2018, 1–43, doi:10.5194/acp-2018-65, 2018.
* * *

---

## Author Response (AR1)

**Diurnal cycle of coastal anthropogenic pollutant transport over southern West Africa during the DACCIWA campaign**

Adrien Deroubaix, Laurent Menut, Cyrille Flamant, Joel Brito, Cyrielle Denjean, Volker Dreiling, Andreas Fink, Corinne Jambert, Norbert Kalthoff, Peter Knippertz, Russ Ladkin, Sylvain Mailler, Marlon Maranan, Federica Pacifico, Bruno Piguet, Guillaume Siour, Solène Turquety

Dear Editor,

We thank you and the referees for their reports. We propose in the following answers along with the appropriate corrections of the manuscript.

**1 Report 1**

**1.1 General Description**

The authors use surface and aircraft observations, radiosondes, satellite observations, and a model to determine the diurnal cycles of NOx and CO at a short-term monitoring site in southern West Africa and assess the contribution of individual sources and synoptic-scale meteorology to this diurnal variability. The figures are clearly presented and the model tracer simulation experiment is informative of the varying contribution of pollution from cities and other sources to pollution at the monitoring site. The content is appropriate for ACP, but there is limited context for the relevance of the results for other time periods and locations in southern West Africa and the implications of findings for future air quality due to increasing urbanisation and changes in biomass burning (Andela et al., 2014). It is for this reason that I'm hesitant to accept as is and highly recommend that the authors strengthen the relevance of the paper by addressing its limited scope.

We acknowledge the reviewer for her/his constructive comments. About the relevance of the paper for other locations or other time periods, we added sentences to state more precisely this important point. The following sentence was added at the end of the introduction, p.3 l.23: *'This study is focused on one specific period and location. However, the conclusions are representative of a longer time period as meteorological conditions at the coast during the so-called monsoon post-onset period were found to be quite stable for several weeks (Knippertz et al., 2017). Spatially, the results are directly representative of the studied region only, the main goal being to estimate the influence of the emissions from four coastal cities on the atmospheric composition in the lower troposphere over inland Benin. Given the broad southwesterly monsoon flow in the region, a similar transport from coastal pollution inland will likely be found along the most of the Guinea Coast.'*

**1.2 General Comments**

- How does the paper fit within the context of other findings from analysing observations and model output during DACCIWA, and also enhance understanding of the region since the AMMA campaign?

  The main goals of the two projects are rather different. In the introduction, we have added the sentence: *'AMMA was focused on the Sahelian region combining a multi-scale approach to better characterize the interactions between atmosphere, land and ocean during the monsoon (Janicot et al., 2008; Redelsperger et al., 2006), whereas the DACCIWA project is dedicated to the interactions between aerosols, clouds and radiation along the highly urbanized coastline of the Gulf of Guinea (Knippertz et al., 2015).'*.
  In this context, this paper aims at focusing on the transport of pollutants from coastal cities to a remote inland site. We have shown that there is a specific time of the day (16:00 to 03:00 UTC) for the transport of pollutants from the coast to the north, which is in agreement with observations and other modeling studies by Adler et al. (2017) and Deetz et al. (2018).

- There are many typos that can be eliminated with a careful reread (extents on P2, L5; 21th on P2, L9; acquiered on P4, L18 etc.).

  The new manuscript has been carefully checked and corrected.

- Please fix incorrect order of in-text citations when there is more than one article in the same year by the same first author (P2, L14; P2, L27 etc.).

  The citations are automatically managed by Latex/bibtex package of the Journal. It has been fixed in the new version and it will have to be control at the proof stage.

- Why does the study focus on Savè, beyond logistics? Is there rapid population increase? Is it an ideal location to understand synoptic scale meteorological patterns in southern West Africa? This could be better justified.

  The study focuses on Savè for two reasons: this is one of the three super-site of the DACCIWA project, and it is the most suited of the three sites to analyze the pollution transport from coast to inland areas because of the low local emissions. As a super-site, a lot of original measurements have been made of air quality issues often affecting smaller inland cities during the campaign. This site is representative of inland air quality, often under the plume of coastal cities.
  This was reworded in the Introduction: the sentence: 'The super-site of Savè serves as a representative location of inland air quality.' has been modified to: 'The super-site of Savè is a representative location to assess the impact of pollution transport from the coast on the air quality of remote inland cities characterized by low local emissions.'
  In Section 2.2 (part Ground based station) a paragraph has been added: 'Three super-sites have been implemented in the framework of the DACCIWA project in Kumasi (Ghana), Ile-Ife (Nigeria) and Savè (Benin). Unlike the two others, Savè is representative of transport-related air quality issues affecting small cities, characterized by low local emissions, downstream of large coastal cities. It is ideal in that the terrain is very flat with no orographically induced circulation impacting the monsoonal flow. Thus to study NOx and CO from coastal urbanized areas, this rural environment is well suited.'
  Moreover, in the Conclusions, a sentence has been added: 'We analyzed pollution transport from the main urban emission centres of the Guinea Coast (Abidjan, Accra, Lomé, Cotonou and Lagos) at the super-site of Savè in order to assess the impact on the air quality of remote inland cities characterized by low local emissions.'

- Only 7 days of observations are considered. Can we draw conclusions about an extended time period based on this brief analysis period? And if so, what time period? The full year? The entire monsoon period? The onset period only?

  Of course, knowing the meteorology in this region, the study is not representative of the whole year. These 7 days are representative of a longer period, mostly the post-onset period defined from 22 June to 20 July by Knippertz et al. (2017). The monsoon flow was not very variable during this period, at least not in direction, as it has been shown by Kalthoff et al. (2018). Moreover, there is a case of long-range transport of biomass burning aerosol from Central Africa during these seven days (Flamant et al., 2018b). This is now better explained in the introduction with the addition of the sentence: 'These seven days are representative the onset period defined from 22 June to 20 July by Knippertz et al. (2017) because of the quite stable wind conditions(Kalthoff et al., 2018). '

- Absent from the study are measurements and/or a discussion of non-methane volatile organic compounds (NMVOCs) and aerosol mass concentrations and composition. Emissions of these are high from local sources and from distant biomass burning (Liousse et al., 2014; Marais et al., 2016; Janicot et al., 2008; Reeves et al., 2010). Can the same conclusions about sources and diurnal variability be drawn about these? Were these measurements made during DACCIWA on the aircrafts flown? If so, do these offer any utility in understanding these pollutants or confirming similar diurnal behavior? If not, are there other studies that could be referenced to assess the sources of these pollutants and the implications for air quality?

  The goal of the paper is not to extensively study all chemical compounds. We have selected $NO_x$ and CO for the sake of simplicity. These two species are enough for the topic of the paper: $NO_x$ is representative of anthropogenic emissions, whereas CO is representative of both anthropogenic and fires emissions. Of course, the addition of VOC and aerosol compositions could be very interesting, but it could blur our message, more focused on meteorology and transport. The chemical concentrations presented in this paper are used to demonstrate that there is a clear pollution transport diurnal cycle associated with each of the four coastal cities on the remote site of Savè.

- Change MODIS-AOD to MODIS AOD throughout.

  This has been changed.

- The figures are presented out of sequence in the text (e.g. Figures 11 and 12 are mentioned before Figure 3). Please reorder the figures so that these are introduced sequentially.

  This is corrected in the new version. It is a problem coming from the compatibility between different Latex's packages.

- Include a link to the MODIS AOD product used in this work (preferably doi, otherwise URL).

  DOIs have been included in the new version.

- Shorten the conclusion to the main findings of the paper, rather than providing a detailed overview of the methods, results, and outcomes.

  The conclusion was rewritten to focus on the main results only.

**1.3 Specific Comments**

- P3, L12: Consider instead referencing a relevant publication from AMMA or from analysis of MOZAIC vertical profiles that finds influence of Central African biomass burning on atmospheric composition in southern West Africa.

  The citation of an article in preparation has been changed to Reeves et al. (2010).

- Table 1: The table title says 2015 population but from a 2011 Revision. Are these projected estimates? Why not use a recent revision? No reference is provided for the 2011 Revision, so it's not possible to assess what the 2015 population is when it was revised in 2011.

  The reference and the url have been included in the new version as a citation (United Nations report, 2011).

- Section 2.2: Are the dashed indented lines headers? These are unnecessary. Instead integrate these in the relevant paragraph and make clear in the first sentence what the paragraph is about.

  We have followed the reviewer' suggestion.

- P5, L9: How frequently are the radiosondes launched during the time period considered in this study?

  In Savè, radiosondes were launched every 1.5 to 3 hours. It has been added in the new version.

- P5, L12-17: How are biomass burning layers identified with a total column AOD measurement?

  The reviewer is right, it is not possible to characterize aerosol vertical layers with an AOD product. The sentence 'We analyze the spatial extent of BB layers from satellite observations' has been changed to 'We analyze the horizontal spatial extent of the main aerosol plumes from satellite observations'.

- P5, L20: How accurate is CALIOP at distinguishing aerosol types?

  As in the case of this study, when remotely sensed aerosols have very different optical properties (dust, biomass burning, sea salt), the CALIOP aerosol classification is robust. This is more thoroughly discussed in the following publication (also made in our team): Menut, L., Flamant, C., Turquety, S., Deroubaix, A., Chazette, P., and Meynadier, R.: Impact of biomass burning on pollutant surface concentrations in megacities of the Gulf of Guinea , Atmos. Chem. Phys., 18, 2687-2707, https://doi.org/10.5194/acp-18-2687-2018, 2018. This reference and an explanation was added in the Section 2.2: 'In order to identify the altitude of aerosols together with their speciation, we use the space-borne Cloud-Aerosol Lidar with Orthogonal Polarization (CALIOP) aerosol type classification (Winker et al., 2009). This classification is suited and accurate to distinguish homogenous aerosol plumes with different optical properties such as sea salt, dust and BB (e.g. Menut et al., 2018).'.

- P5, L28-29: Knippertz et al. (2017) isn't the seminal paper on defining the monsoon onset period. Do the authors mean to say that the monsoon onset period specific to DACCIWA (2016) is 22 June to 20 July?

  This article deals only with the 2016 WAM. Our studied period is the post-onset period for the WAM 2016. This is now specified in Section 2.3: *'The studied period (1 to 7 July) is entirely included in the 2016 WAM post-onset phase, which has been defined from 22 June to 20 July 2016 by Knippertz et al. (2017).'*.

- P6, L23-30: Elaborate on the pseudo-anthropogenic emissions that are imposed. Are these based on an existing inventory? How much is emitted per one million inhabitants? Does the imposed magnitude of emissions matter?

  The expression *'pseudo-anthropogenic emissions'* is misleading. We want to scale the tracer emissions to the size of the agglomeration (Section 2.1), as it has been done by Flamant et al. (2018a). This paragraph has been reworded: *'In this study, the model is used in its tracer version and there is no atmospheric chemistry. We choose to release passive gaseous tracers in the simulation, because we want to analyze only their transport (no chemistry, no deposition) caused by the monsoonal flow. Since we want to distinguish the relative contribution of several coastal cities to the pollution further inland, we designed a first experiment for which we impose the tracer emissions at specific urbanized locations: Abidjan (Ivory Coast), Accra (Ghana), Lomé (Togo), Cotonou (Benin), Lagos (Nigeria) (cf. Table 1 for coordinates). Specific tracers are emitted for a given city in order to distinguish their relative contributions at inland locations. Thus the tracer emissions occur in a single grid cell corresponding to the center of each city.'*

- P7, L9-11: Was there any rain during 1-7 July that would cause aerosols to wet deposit and so affect interpretation of output from a model that does not include sinks?

  The following sentence has been included in Section 2.3.2: *'The period is not associated with widespread rain. At Savè, there were only some small precipitation events (Kalthoff et al., 2018). We focus our analysis on gaseous species but we suppose similar transport patterns for aerosol and gaseous pollutants because of the constant monsoon flow blowing over SWA during our studied period.'*

- P7, L26-28: This transport pattern has already been observed during AMMA and before that using MOZAIC aircraft campaign observations. Consider including these studies too to reinforce that this work builds on previous research and campaigns.

  We have included two articles from the AMMA campaign results. These sentences have been modified: *'The origin of the high AOD over the Gulf of Guinea is well known. This is the BB layer coming from Central Africa, where intense vegetation fires occur during this season (Giglio et al., 2006) with increasing trends over the period 2001-2012 (Andela and Van Der Werf, 2014). Part of this pollution is transported over the Gulf of Guinea and the BB plume reaches the Guinea Coast was seen during the AMMA campaign (Sauvage et al., 2005; Reeves et al., 2010). The BB pollutant concentrations observed along the Guinea Coast depend on the synoptic wind patterns (Menut et al., 2018).'*

- Figure 3: How representative is the site of the coincident 2 km x 2 km model gridsquare?

  In the legend of Figure 3, we have added the sentence: *'WRF-derived variables are interpolated to the radiosonde positions'*.

- Section 3.2: 'From research aircraft' and 'From radiosondes' are not complete headers. Consider revising.

  The titles have been revised as follows: Section 3.2 *'Vertical layers in the lowermost troposphere'*; and Subsections 3.2.1 *'Identification from radiosondes'* as well as 3.2.2 *'Identification from research aircraft'*.

- P11, L1: Why define this "feature G". It's not used again and doesn't seem to appear in any of the figures.

  This was unclear and unnecessary. 'Feature G' refers to a feature described in Knippertz et al. (2017). The parenthesis *'(described as feature G in this study)'* has been removed.

- P12, L30-35: Does interference in the NO2 measurement from dissociation of NOx reservoir compounds (Reed et al., 2016) impact the interpretation of NO2 and NO2/NO diurnal variability?

  The interference described in Reed et al. (2016) is significant for low $NO_x$ concentration (about 10 ppt). Given the relatively high concentrations of $NO_x$ (greater than 0.2 ppb) over our domain, we can assume a low effect of this process on our results.

- P13, L27-29: This result isn't surprising, as the monsoon flow is prevailing southwesterly and so transport from Lagos isn't expected. It would be more noteworthy if this did occur.

  We agree with the reviewer. The sentence *' It is worthy of note that Lagos tracers do not reach Savè, although emissions from Lagos are greater than from the other cities.'* has been replaced by *'Lagos tracers do not reach Savè because of the southwesterly monsoon flow.'*.

- P14, L1: What is the implication on future air quality that the Cotonou plume affects Savé?

  The anthropogenic emissions of the agglomeration of Cotonou are supposed to rise because the population is quickly growing. Beyond this point, our results have shown that there is a specific time of the day (16:00 to 02:00 UTC) for the transport of pollutants from the coast toward the north. There are implications for inland air quality over SWA, human health as well as radiative transfer and the diurnal cycle of low-level clouds.
  We have added a sentence in the Conclusions: *'There is a specific time of the day (16:00 to 02:00 UTC) for the transport of pollutants from the coast toward the north, which affects inland air quality over SWA, human health as well as radiative transfer and low-level cloud diurnal aspects.'*

- Videos: Consider adding annotations or narration to the videos to guide the viewer. Also consider adding time stamps in the text to point the reader to these specific features in the videos.

  We have followed the reviewer suggestion by adding time stamps in the text of Section 4.3: *'From the wind patterns, we note that the coastal front is present from 09:00 to 15:00 UTC from the coast up to ≈ 7° N. It leads to the accumulation of URB pollutants. This period is referred to 'Daytime drying' by Deetz et al. (2018). From 16:00 to 02:00 UTC (respectively from 03:00 to 08:00 UTC), the meridional wind increases (resp. decreases) in the PBL and URB pollutants are mostly transported northward within the PBL. This period is referred to 'Atlantic inflow' (resp. 'Moist morning') by Deetz et al. (2018).*
  *During the 'Daytime drying' period, we notice that URB and BB tracers accumulate along the coastline in the PBL (on 5 and 6 July from 11:00 to 15:00 UTC). When dry convection stops (at 16:00 UTC), wind speed quickly increases with a stronger northward component. BB and URB tracers are simultaneously transported northward from the coast. From 16:00 to 02:00 UTC, the front moves towards the north, and the mixture of BB and URB tracers is advected accordingly. A similar diurnal evolution of BB and URB transport is simulated on 5 and 6 July. '*

**1.4 References**

- Andela et al., doi:10.1038/nclimate2313, 2014.
- Janicot et al., doi:10.5194/angeo-26-2569-2008, 2008.
- Knippertz et al., doi:10.5194/acp-17-10893-2017, 2017.
- Liousse et al., doi:10.1088/1748-9326/9/3/035003, 2014.
- Marais et al., doi:10.1021/acs.est.6b02602, 2016.
- Reed et al., doi:10.5194/acp-16-4707-2016, 2016.
- Reeves et al., doi:10.5194/acp-10-7575-2010, 2010.

  All these references are now included; except Reed et al. (2016) because the interference described in this article is significant for low $NO_x$ concentration (about 10 ppt).

**2 Report 2**

**2.1 General Description**

The authors used ground-based and aircraft measurements of meteorological and chemical (CO, NO, NO2) variables collected during DACCIWA campaign to understand the transport pathway of biomass burning and

coastal urban emissions in SWA. To further investigate this, they designed and used two experiments using WRF-CHIMERE (in tracer mode). First, they thoroughly studied the model performance and its limitations in capturing the transport. Next, they used the tracer experiments to assess the contribution of urban and biomass burning pollutions in the region. The manuscript is well written, and I recommend it for ACP after fixing the typos and addressing some issues listed below.

We thank the reviewer for her/his positive comments.

**2.2 General Comments**

- There are mistakes in the order of references and format of the references in the introduction. Please double check the references.

  It has been entirely revised.

- UTC and local time description (morning/evening) were used interchangeably. Please provide information on the time zone and sunset/sunrise time.

  This information have been added in Section 2.2: 'Our studied domain is located in the Greenwich Mean Time (GMT). Hence local time is the same as UTC. During the aircraft campaign period, sunrise occurred arund 06:00 UTC and sunset around 18:00 UTC.'

- The section on population can be shortened as some parts in introduction and 2.1 overlaps.

  The section 2.1 has been reworded following the other reviewer's comment.

- How does the result compare with the AMMA campaign?

  During the AMMA aircraft campaign, most aircraft operations took place over the Sahel, whereas the DACCIWA aircraft campaign was focused on the Guinea Coast. Moreover, the goal of the AMMA project was to better characterize the interactions between atmosphere, land and ocean during the monsoon (Janicot et al., 2008; Redelsperger et al., 2006), whereas the DACCIWA project was dedicated to the interactions between aerosols, clouds and radiation along the highly urbanized coastline of the Guinea Coast (Knippertz et al., 2017).
  In the introduction, we have added the sentence: 'AMMA was focused on the Sahelian region combining a multi-scale approach to better characterize the interactions between atmosphere, land and ocean during the monsoon (Janicot et al., 2008; Redelsperger et al., 2006), whereas the DACCIWA project is dedicated to the interactions between aerosols, clouds and radiation along the highly urbanized coastline of the Gulf of Guinea (Knippertz et al., 2017)'.

- The order of the figures is not correct. Please use supplement material if necessary.

  Supplement material has been added and the order of the figure has been corrected in the new version.

- In section 3.2.2. the discussion on model performance in capturing wind and RH is difficult to follow. Please either discuss layer-by-layer (starting from the lowest layer) or variable by variable.

  We have followed the reviewer's suggestion and this section has been reworded:
  'From the surface to 1 km amsl, the modeled wind speed and direction match well with the observations (absolute bias lower than 0.2 $m.s^{-1}$ and $4^o$, respectively). The observed distribution of the monsoon wind is captured by the model up to 1 km amsl (inter-quartile range $3.80 - 6.47$ $m.s^{-1}$ for the observations and $3.98 - 6.80$ $m.s^{-1}$ in the simulation). The modeled distribution of RH shows a dry bias in the monsoonal flow (of -6 %).
  In the directional shear layer, from 1 to 2 km amsl, observed and modeled wind speed distributions are narrower than in the monsoon layer (inter-quartile ranges being $2.10 - 4.04$ $m.s^{-1}$ and $1.88 - 3.94$ $m.s^{-1}$ respectively), showing that this layer is well defined over the domain. The modeled wind direction is in good agreement with observations (relative bias lower than 1 %), although with a wider distribution (observed inter-quartile range $191.17 - 285.29$ $^o$ and modeled $214.17 - 237.67^o$) than in the first kilometer (observed inter-quartile range $220.08 - 244.41^o$ and modeled $207.29 - 272.28^o$).
  In the main easterly layer, from 2 to 4 km amsl, the observed distribution of wind speed is wider than at lower altitudes, which is in good agreement with the modeled distribution. Observed and modeled RH and

*wind direction distributions are also consistent.*
*In conclusion, during daytime, the monsoon layer is reproduced with a dry bias. The low (relative) biases of wind speed (+4 %) and direction (-2 %) in this layer are of prime importance to accurately model the URB transport.'*

- In section 3.2.2. How does the result compare with radiosondes results?

  We use the vertical layers identified with radiosonde observations to analyze the aircraft observations in these layers. Aircraft and radiosondes provide complementary information on horizontal and vertical variability, respectively. It is thus difficult to compare these observations.

- Estimating PBL height from radiosonde measurements (separating day and night measurements) and comparing with model PBL height can be beneficial for the discussion in section 3.3
- I suggest adding more details on the PBL height and its influence on the concentration of different constituent, especially during nighttime.

  We took on the reviewer's comments and started looking into this. However, determining the PBL height from the radiosoundings at the coast proved to be difficult because of the presence of multiple low-level cloud layers as well as the relatively weak associated temperature inversions. The results were not satisfactory, and hence we did not pursue this further. Moreover, Section 3.3 focuses on daytime observations acquired by the aircraft. For the horizontal pollution transport, which is the main concern of the article, our goal is to capture the wind speed and direction. With the numerical tracers, we use an arbitrary unit and no chemistry, thus the dilution volume is not the main driver of the variability.

**2.3 Specific Comments**

- P3-L1-5: I suggest adding more references on the source attribution to provide a bigger picture. What is the contribution of these sectors in other regions or bigger domains? For example, compare with Sobhani et al., 2018; Yang et al., 2017; Kulkarni et al., 2015

  These references have been added (Kulkarni et al., 2015; Yang et al., 2017; Sobhani et al., 2018) in the introduction as follows: *'In order to better distinguish the contributions from different sources to background concentrations, additional studies are needed focusing on Africa as it has been done for other regions (e.g. Kulkarni et al., 2015; Yang et al., 2017; Sobhani et al., 2018).'*

- P3-L2: What year? 2006?

  Yes, this study was done using datasets acquired during the AMMA campaign in 2006. This sentence has been reworded: *'Using WRF-CHIMERE numerical simulations of the WAM during the African Monsoon Multidisciplinary Analyses (AMMA) campaign period from May to July 2006, Deroubaix et al. (2018) quantified the relative contributions of anthropogenic and biomass burning sources to carbon monoxide (CO) concentrations over SWA, which in July 2006 was about 25 % local anthropogenic and 50 % biomass burning from Central Africa. '*

- P3-L12: "Haslett et al., in preparation" I don't think this is an acceptable format and it is not mentioned in the References section.

  It has been replaced by Reeves et al. (2010).

- P3-L18: "1-7 July" Please add 2016.

  The year 2016 has been added.

- P4-L9: the Lome population stated in the text does not match Table 1.

  We thank the reviewer for picking this up. It has been corrected.

- P4-L10 and the next paragraph: Information given on population of Cotonou is confusing. Is it necessary to state 1,086,00 inhabitant and then change it in the next paragraph?

  With the tracer experiment, we scale the tracer emissions to the size of the population in each city. We have used the estimation of the department of social and economic affairs of United Nations (United Nations report, 2011), except for Cotonou because we were surprised by the fact that Lomé was estimated

bigger than Cotonou. In Section 2.1, we justify this assumption but it was not well written. It has been improved in the new version of the article: *'We focus on six locations, five major urban agglomeration of the Guinean coastal region, and one small town, Savè, which is 185 km north of Cotonou (Benin). Table 1 shows the coordinates and the population of the urban agglomerations studied. For Abidjan, Accra, Lomé and Lagos, we present estimations for the year 2015 of the department of social and economic affairs of United Nations (United Nations report, 2011). In this report, these cities are associated with large administrative areas in contrast to Cotonou.*

*Comparing Lomé and Cotonou, Lomé has a large administrative area while Cotonou is a city with a very high population density over a small area. The population of Lomé is about 839,000 inhabitants in the city according to the General Population and Habitat Census of Togo ( DGSCN report, 2016) and about 1,830,000 inhabitants in the administrative state (United Nations report, 2011). Cotonou city is estimated by the World Urbanization Prospects report to have about 1,086,000 inhabitants (United Nations report, 2011).*

*According to the General Population and Habitat Census of Benin (INSAE report, 2015), the population of Cotonou only slightly increased by 2.09 % over the period 2002-2013 because of the limited possibility of expansion. They note that along the shores of Lake Nokou the population has increased rapidly, thus forming an agglomeration of 2,194,000 inhabitants, calculated as the sum of the Cotonou district and several cities of the Atlantique district (Abomey-Calavi and So-Ava) and of the Ouémé district (Seme Kpodji, Porto Novo, Avrankou and Akpro-Misserete). '*

- P42-Table 2: Please provide coordinates of the ground sites.

  The ground site coordinates have been provided in Table 2.

- P5-L9: What is the local time?

  In this region and at this time of year, UTC is the same as local time. Following the previous reviewer's comment, indications on time zone have been added in Section 2.2.

- P5-section 2.3: Please add a figure of outer domain.

  The introduction of section 2.3 have been modified to include this information: *'The WRF-CHIMERE simulations presented in this study have a similar set-up to those used by Deroubaix et al. (2018). Both models are run offline in a nested configuration on the same grids with two domains: a regional domain (10 km×10 km, extending from $1^o$ S to $14^o$ N and from $11^o$ W to $11^o$ E) and a high-resolution coastal domain (2 km×2 km). The simulations over the regional domain are started on 1 June 2016. In the following, we present only results modeled over the high-resolution domain (Figure 1). The simulated period over the high-resolution domain (1 to 7 July) is entirely included in the 2016 WAM post-onset phase, which has been defined from 22 June to 20 July 2016 by Knippertz et al. (2017). '.*

- P6-L2: How many layers below 1 km?

  This is an important information regarding the goal of our study. It has been added in the introduction of section 2.3.1: *'The domains have a constant horizontal resolution with 32 vertical levels from the surface to 50 hPa, including about 10 vertical levels below 1 km amsl.'*

- P6-L5: Please reference GFS data.

  The DOI is now mentioned: *'Global meteorological fields are taken from the US Global Forecast System (operational final analyses) produced by the National Center for Environmental Prediction (ds083.3 dataset DOI: https://10.5065/D65Q4T4Z)'*

- P6-L5: Did you use GFS data only for nudging or for meteorological initial and boundary conditions as well? Please clarify in the text.

  The GFS dataset has been used for both. It is now clearly stated.

- P6-L14: Did you use cumulus parametrization for both domains?

  The reviewer is right. We use a cumulus parametrization only for the regional domain. This information is now specified in this sentence: *'for the high-resolution domain, convective precipitation is explicitly calculated and not parametrized'.*

- P6-L27: What time do you start releasing the tracers? Have you allowed for spin up in the model before releasing the tracers?

  We use a one month spin-up over the regional domain. This information was missing and has been added in the introduction of Section 2.3: *'The simulation over the regional domain are started on 1 June.'*

- P6-L28: "grid cell of each city". One grid cell? Given the high resolution of your inner domain, did you release tracers from one grid cell or from a region (city)? Please clarify in the text.

  We release the tracer in a single grid cell for each city. This assumption leads to pollution plumes thinner as the reality. But, given the strong urbanization of the Guinean coastal region, there are anthropogenic emissions almost continuously from Abidjan to Lagos. As we want to distinguish the contribution of the major agglomeration, we release the tracers from a single grid cell.
  The sentence *'The tracer emissions occur in the grid cell of each city.'* has been changed as: *'Specific tracers are emitted for a given city in order to distinguish their relative contributions at inland locations. Thus the tracer emissions occur in a single grid cell of each city corresponding to the center.'*.

- P7-L1: How much tracer did you release? Please clarify how you "reproduce the BB layer observed with MODIS" considering that the tracers are gaseous.

  The quantity of tracer is not important because we use an arbitrary unit. The expression *'reproduce the BB layer observed with MODIS'* meant that we reproduce the spatial extent of the aerosol pattern seen with MODIS AOD and supposed to be associated with a BB layer (Section 3.1).
  The sentence *'In order to analyze the interactions between URB emissions and the BB layer, we reproduce the BB layer releasing passive gaseous tracers from 5 July at 00:00 UTC to 6 July at 23:00 UTC with a spatial horizontal extent from 1º W to 2º E at 4.5º N, and the altitude as suggested by satellite observations (Figure A1) at ≈ 1.5 km amsl.'* has been reworded: *'In order to analyze the interactions between URB emissions and the BB layer, we use the information on the spatial characteristics and vertical extent of the BB layer derived from MODIS and CALIPSO observations (Section 3.1) by releasing passive gaseous tracers with a spatial horizontal extent from 1º W to 2º E at 4.5º N and with an altitude of ≈ 1.5 km amsl from 5 July at 00:00 UTC to 6 July at 23:00 UTC (cf. Table 3).'*.

- P9-L16: "Section 3.2" Do you mean Section 3.2.1?

  The reviewer is right. It has been corrected.

- P9-L18: Please comment on RH in layer (ii) to be consistent.

  It has been commented in the new version: *'(ii) between 1 to 2 km amsl, there is a directional shear layer associated with high RH > 90 % and with low wind speed and changing wind direction;'*.

- P9-L24: "the first kilometer" is this the monsoon layer? Please be consistent.

  In order to be consistent, we have replaced 'the first kilometer' by 'the monsoon layer'.

- P9-L27: "wind speed is wider than at lower altitudes". Remove "than".

  It has been corrected.

- P10-L18: Did you compare measured and modeled RH? This can help to better understand the model performance in capturing PBL height and (maybe) clouds.

  Observed and modeled RH are compared in Section 3.2. We decided that the comparison of observed and modeled RH is not important for Section 3.3 because we want especially to validate the wind speed and direction in order to accurately reproduce the pollutant transport, which is the focus of our study.

- P11-L1: What is "feature G"?

  This was unclear and unnecessary. 'Feature G' refers to a feature described in Knippertz et al. (2017). The parenthesis *'(described as feature G in this study)'* has been removed.

- P11-L22: NO and NO2 have a short lifetime and are not trace gases.

  That is true. We have replaced *'this section investigates the trace gas concentrations'* by *'this section investigates CO and $NO_x$ concentrations'*.

- P29-Figure 5: Do thick marks indicate 00Z?

  The vertical dashed lines (and the corresponding tick marks) indicate periods of 6 hours starting at 00:00 UTC (00Z). It has been specified in the caption of Figure 5: *'The vertical dashed lines indicate periods of 6 hours starting at 00:00 UTC.'*.

- P12-L14-15: Increase in evening NO2/NO can be due to lower PBL (higher NO2 concentration) and reduction in NO by reacting with O3.

  Close to the sources of incomplete combustion, there is always production of NO. We interpret the $NO_2$/NO ratio together with the NO concentration in order to confirm that there is no local production. The sentence *'We note every day a sharp increase in the evening ($NO_2$/NO > 15), which suggests transported pollutants.'* has been modified such as: *'We note every day a sharp increase from 18:00 to 00:00 UTC ($NO_2$/NO > 15) associated with low NO concentrations (NO < 0.2 ppb), which suggests transported pollutants.'*

- P12-L20: On July 5th concentration of both CO and NOx (more) increased in the evening thus resulted in a decrease in CO/NOx ratio and increase in NO2/NO. Maybe a mixture of BB and URB was transported to Save?

  We have highlighted this fact in the manuscript. It shows that there is no increase of CO without $NO_x$ increase, thus if the BB layer reach Savè, it is mixed with urban pollution.
  The sentence *'At Savè, the $CO/NO_x$ ratio is not higher after the arrival of the BB layer on 5 July (cf. Section 3.1).'* has been modified such as *'At Savè, the $CO/NO_x$ ratio is not higher after the arrival of the BB layer on 5 July (cf. Section 3.1). There is an increase of CO together with a $NO_x$ increase, thus when the BB layer reaches Savè, it is mixed with urban pollution or transported above the PBL.'*

- P12-L23: no need to start a new paragraph.

  OK

- P12-28-29: Could the nighttime increase in CO (and NO2 in the next paragraph) concentrations be due to lower PBL height? Please comment in the text.
- P12-L34: Again, lower nighttime PBL height can also justify the increase in CO and NO2 concentrations.
- P13-L4: or shorter lifetime of NO2?

  For these three points, we think that it would be the case if there are anthropogenic emissions at night in the vicinity of the measurements. During the entire campaign, we did not note any increase of NO at night. We consider that the super-site of Savè is well suited to study pollution transport from the coast to the North because of the low anthropogenic emissions. This information is now provided with a new paragraph in Section 2.2: *'Three super-sites have been implemented in the framework of the DACCIWA project in Kumasi (Ghana), Ile-Ife (Nigeria) and Savè (Benin). Unlike the two others, Savè is representative of transport-related air quality issues affecting small cities, characterized by low local emissions, downstream of large coastal cities. It is ideal in that the terrain is very flat with no orographically induced circulation impacting the monsoonal flow. Thus to study NOx and CO from coastal urbanized areas, this rural environment is well suited.'*
  We understand the question of the reviewer as a question about the relative quantification of several dynamical processes on surface concentrations increases. The question is: *'Observing CO and NO2 surface concentrations increases, is it due to long-range transport of urban plumes and/or local collapse of the nocturnal boundary layer?'*. The answer is that we have shown with the numerical tracer experiments (Section 4.2) that the long-range transport is important for the surface concentration increases at Savè in the evening and could explain entirely these increases.
  In the past paragraph of Section 4.2, this sentence was added: *'The observed increases in surface concentration may be explained by several processes: long-range transport of the URB plume and/or a local collapse of the nocturnal boundary layer, quickly concentrating locally emitted pollutants. In our case, the dominant effect is the long-range transport, which is mainly associated with the transport of the Cotonou plume. Moreover, during the entire campaign, we did not note any increase of NO at night (Figure A4 and A5).'*.

- P12-L32: What about NO peak at 12UTC?

It was missing. It has been added in the sentence: *'The two small NO increases (from 0.2 to 0.6 ppb) at 12:00 and 19:00 UTC are consistent with the usual time of local activities such as traffic and charcoal cook stoves.'*.

- P13-L13: please reference the figure after "single figure"

  OK

- P13-L16: "northwestward" you mean "southwestward"?

  It has been corrected.

- P13-L21: What time did you start releasing the tracers?

  The sentence *'The first tracer plume that reaches Savè typically around 19:00 UTC is from Cotonou.'* has been modified *'The tracer emissions started on 1 July at 00:00 UTC and the first tracer plume that reaches Savè is from Cotonou in the evening. We can see that tracers emitted from Cotonou reach Savè every day in evening, typically at around 19:00 UTC.'*

**2.4 References**

- Yang, Y., Wang, H., Smith, S. J., Ma, P.-L. and Rasch, P. J.: Source attribution of black carbon and its direct radiative forcing in China, Atmos. Chem. Phys., 17(6), 4319-4336, doi:10.5194/acp-17-4319-2017, 2017.
- Kulkarni, S., et al. "Source sector and region contributions to BC and PM 2.5 in Central Asia." Atmospheric Chemistry and Physics 15.4 (2015): 1683-1705.
- Sobhani, N., Kulkarni, S. and Carmichael, G. R.: Source Sector and Region Contributions to Black Carbon and PM2.5 in the Arctic, Atmos. Chem. Phys. Discuss., 2018, 1-43, doi:10.5194/acp-2018-65, 2018.

  These references have been added to the manuscript.

main goal being to estimate the influence of the emissions from four coastal cities on the atmospheric composition in the lower troposphere over inland Benin. Given the broad southwesterly monsoon flow in the region, a similar transport from coastal pollution inland will likely be found along the most of the Guinea Coast. We shall answer these questions using a synergistic combination of observations and numerical modeling experiments, described in Section 2. Section 3 analyzes the

5   temporal evolution of meteorology and air pollution over a portion of SWA including Ivory Coast, Ghana, Togo, Benin and Nigeria. Then, we focus on Urban Anthropogenic (URB) and long-range Biomass Burning (BB) pollutant transport in Section 4. Conclusions are given in Section 5.

**2   DACCIWA project: observations and modeling**

In the DACCIWA project, there are strong components on both in-situ observations and modeling. Here, we present all studied

10   sites (2.1), observational datasets used (section 2.2), and numerical simulations performed to analyze the pollution transport pathways (section 2.3).

**2.1   Studied sites**

We focus on six locations, five major urban agglomeration of the Guinean  coastal region, and one small town, Savè, which is 185 km north of Cotonou (Benin). Table 1 shows the coordinates and the population of the urban agglomerations

15   studied. For Abidjan, Accra, Lomé and Lagos, we present estimations for the year 2015 of the department of social and economic affairs of United Nations (United Nations report, 2011). In this report, the cities of Abidjan, Accra, Lagos and Lomé are associated with large administrative areas in contrast to Cotonou.

Comparing Lomé and Cotonou, Lomé has a large administrative area while Cotonou is a city with a very high population

20   density over a small area.  The population of Lomé is about 839,000 inhabitants in the city according to General Population and Habitat Census of Togo (DGSCN report, 2016)and about 1,830,000 inhabitants in the administrative state (United Nations report, 2011). Cotonou city is estimated by the World Urbanization Prospects  report to have about 1,086,000 inhabitants (United Nations report, 2011).

25   According to the General Population and Habitat Census of Benin (INSAE report, 2015), the population of Cotonou only slightly increased by 2.09 % over the period 2002-2013 because of the limited possibility of expansion. They note that along the shores of Lake Nokoué the population has increased rapidly, thus forming an agglomeration of 2,194,000 inhabitants, calculated as the sum of the Cotonou district and several cities of the Atlantique district ( Abomey-Calavi and So-Ava) and of the Ouémé district (Seme Kpodji, Porto Novo, Avrankou and Akpro-Misserete).

| City | Country | Latitude | Longitude | Elevation | Number of inhabitants |
|--------|------------|----------|-----------|------------|------------------------|
| Abidjan | Ivory Coast | 5.36°N | 4.00°W | 50 m amsl | **4 923 000** |
| Accra | Ghana | 5.60°N | 0.19°W | 30 m amsl | **3 013 000** |
| Lomé | Togo | 6.17°N | 1.23°E | 10 m amsl | **1 830 000** |
| Cotonou | Benin | 6.36°N | 2.38°E | 10 m amsl | 2 194 000 |
| Savè | Benin | 8.03°N | 2.49°E | 130 m amsl | 87 000 |
| Lagos | Nigeria | 6.49°N | 3.36°E | 10 m amsl | **13 121 000** |

**Table 1.** *Characteristics of the studied cities with: country, latitude, longitude, elevation above mean sea level (amsl), and number of inhabitants of urban agglomerations. The population in bold is given for the year 2015 according to World Urbanization Prospects (The 2011 Revision)report (United Nations report, 2011). National general population and habitat census is used to estimate the population of Cotonou and Savè (2013 Census)(INSAE report, 2015).*

**2.2 Observational datasets**

During the DACCIWA field campaign, several observational platforms were deployed to perform in-situ and remote sensing measurements (Knippertz et al., 2017; Flamant et al., 2018b). In this study, we use datasets acquiered acquired by ground based stations, aircraftsaircraft, radiosondes and satellites. Table 2 gives the main information on each dataset. Figure 1 presents the location of the aircraft flight tracks, and of the stations. Our studied domain is located in the Greenwich Mean Time (GMT). Hence local time is the same as UTC. During the aircraft campaign period, sunrise occurred around 06:00 UTC and sunset around 18:00 UTC.

| Datasets | Platform | Variables | Frequency |
|----------|----------|-----------|-----------|
| Ground based station | Savè super-site (8.03°N, 2.49°E) | $NO_2$, NO | Raw data: 1 hz |
| | operated by KIT-UPS universities | and CO concentrations | Presented: hourly averages |
| Aircraft | ATR-42, Twin Otter, Falcon | Relative humidity | Raw data: 1 hz |
| | operated by SAFIRE, BAS and DLR teams | Wind direction and speed | Presented: 3-min averages |
| Radiosonde | Launch sites: | Wind direction and speed | High resolution 1 hz |
| | Abidjan, Accra, Cotonou, Savè | Relative humidity | Presented: 100 m averages |
| Satellite | MODIS on Terra | AOD (550 nm) | Daily |
| | and Aqua | level 3 (1°x1°) | |

**Table 2.** *Datasets used in this study with acquisition platform, variables and sampling frequency.*

– Ground based station

Three super-sites have been implemented in the framework of the DACCIWA project in Kumasi (Ghana), Ile-Ife (Nigeria) and Savè (Benin). Unlike the two others, Savè is representative of transport-related air quality issues affecting small cities,

[Figure]

**Figure 1.** *Map of the modeling domain (red rectangles) with location of the major cities (red dots), of the Lomé airport and the Savè super-site (white dots). Superimposed are the flight tracks of the three research aircraft during the  1–7 July 2016 period (grey lines). The aircraft flight tracks on 5 July are colored for the German Falcon (blue line), the French ATR-42 (violet line) and the British Twin Otter (yellow line).*

characterized by low local emissions, downstream of large coastal cities. It is ideal in that the terrain is very flat with no orographically induced circulation impacting the monsoonal flow. Thus to study NOx and CO from coastal urbanized areas, this rural environment is well suited.

At Savè, the Karlsruhe Institute of Technology (KIT) and the Paul Sabatier University (UPS) have set-up
5  meteorological and atmospheric composition measurements. UPS installed a chemical analyzer (ThermoEnvironment Instrument) which measured $NO_2$, NO and CO surface concentrations (Pacifico et al., 2018; Kalthoff et al., 2018). Raw observations acquired at 10 s are averaged hourly. The detection limit of the instrument is 0.05 ppb for $NO_2$ and NO, and 12 ppb for CO (Derrien and Bezombes, 2016). The measurements site is upwind of the Savè city when the wind corresponds to the monsoon flow (SW sector). On 3 July from 18:00 to 21:00 UTC, the wind direction was shifted, which
10  corresponds to local pollution. This period has been removed from the analysis.

    –

The DACCIWA aircraft campaign took place during the period 25 June - 14 July 2016 and was based at the Lomé (Togo) airport (Flamant et al., 2018b). Three research aircraft were involved: a Twin Otter operated by the British Antarctic Survey (BAS), an ATR-42 operated by the French "Service des Avions Français Instrumentés pour la Recherche en Environment"
15  (SAFIRE), and a Falcon operated by the German "Deutsches Zentrum für Luft und Raummfahrt" (DLR). We base our study on three variables, namely relative humidity, wind direction and wind speed, which are measured by core meteorological

instrumentation. The flights trajectories used are depicted in Figure 1. For the three aircraft, raw observations acquired at 1 Hz are averaged over 3-minute time steps.

–

The DACCIWA project included a large radiosonde component with locations carefully chosen building on the AMMA radiosonde campaign experiences (Lothon et al., 2008; Parker et al., 2008; Fink et al., 2011; Schuster et al., 2013). We use radiosondes launched from four locations: Abidjan, Accra, Cotonou and Savè (*cf.* Table 1). There were four releases per day at around 00:00 UTC, 06:00 UTC, 12:00 UTC and 18:00 UTC. In Savè, more radiosondes were launched every 1.5 to 3 hours at the super-site during the Intensive Observation Period of  1–7 July 2016  (Kalthoff et al., 2018).

–

We analyze the horizontal spatial extent of  the main aerosol plumes from satellite observations of Aerosol Optical Depth (AOD) at 550 nm made by MODIS (MODerate-resolution Imaging Spectroradiometer) on both the Aqua platform (MYD08-D3-6 dataset DOI: https://10.5067/MODIS/MYD08_D3.061, passing over the studied region at 13:30 UTC) and the Terra platform (MOD08-D3-6 dataset DOI: https://10.5067/MODIS/MOD08_D3.061, passing over the studied region at 10:30 UTC). Daily MODIS AOD averages at 1° resolution have been calculated from the Collection 6 combined product of the Dark-Target retrieval available over oceans or non-bright continental surface, and the Deep-Blue retrieval available over deserts (Hsu et al., 2013; Sayer et al., 2013, 2014).

–

In order to identify the altitude of aerosols together with their speciation, we use the space-borne Cloud-Aerosol Lidar with Orthogonal Polarization (CALIOP)  aerosol type classification (Winker et al., 2009). This classification is suited and accurate to distinguish homogenous aerosol plumes with different optical properties such as sea salt, dust and BB (e.g. Menut et al., 2018). The CALIOP cross-sections are very useful since this is a realistic way to have an instantaneous evaluation of the aerosol layer altitudes, with their type depending on backscattered optical measurements,  (e.g. Winker et al., 2013). Data are available on https://www-calipso.larc.nasa.gov/products/lidar/browse_images.

**2.3 Numerical modeling by WRF-CHIMERE models**

The WRF-CHIMERE simulations presented in this study have a similar set-up to those used by Deroubaix et al. (2018). Both models are run offline in a nested configuration on the same grids  with two domains: a regional domain (10 km×10 km, extending from 1°S to 14°N and from 11°W to 11°E) and a high-resolution coastal domain (2 km×2 km). The simulations over the regional domain are started on 1 June 2006. In the following, we present only results modeled over the high-resolution domain  (Figure 1). The simulated period over the high-resolution domain (1 to 7 July) is entirely included in  the 2016 WAM post-onset phase, which has been defined from 22 June to 20 July 2016 by Knippertz et al. (2017).

**2.3.1 Meteorological fields from the WRF model**

Meteorological variables are modeled with the regional non-hydrostatic WRF model (version 3.7.1, Skamarock and Klemp, 2008). The domains have a constant horizontal resolution with 32 vertical levels from the surface to 50 hPa, including about 10 vertical levels below 1 km amsl. We use a 2-way nesting for the communication between different domains.

5  Global meteorological fields are taken from the US Global Forecast System (operational final analyses) produced by the National Center for Environmental Prediction (ds083.3 dataset DOI: https://10.5065/D65Q4T4Z). These fields are used to provide meteorological initial and boundary conditions, and to nudge hourly fields of pressure, temperature, humidity and wind in the WRF simulations, with spectral nudging, which has been evaluated for regional models by von Storch et al. (2000). In order to enable the PBL variability to be resolved by WRF, low frequency spectral nudging is used only above 850 hPa.

10  The WRF model set-up is as follows: The microphysics scheme is the Single Moment-6 class microphysics scheme (WSM6), the radiation scheme is the Rapid Radiative Transfer Model for General Circulation Models (RRTMG) with the Monte-Carlo Independent Column Approximation (McICA) method of random cloud overlap from Mlawer et al. (1997), the PBL physics are computed using the Yonsei University scheme (Hong et al., 2006), the cumulus parametrization is the ensemble Grell-Dévényi scheme  (for the high-resolution domain, convective precipitation is explicitly calculated and not parametrized), the
15  surface layer scheme is the Carlson-Boland viscous sub-layer and the surface physics is calculated with the 'Noah' Land Surface Model scheme with four soil temperatures and moisture layers (Ek et al., 2003). This set-up has already been used by Deroubaix et al. (2018) because it allows to reproduce a satisfactory diurnal cycle of wind speed over SWA according to Schuster et al. (2013).

**2.3.2 Gaseous tracers transport from the CHIMERE model**

20  CHIMERE is a regional chemistry-transport model (version 2017), fully described in Menut et al. (2013) and Mailler et al. (2016). The 32 vertical levels of the WRF model are projected onto the 20 levels for CHIMERE from the surface and up to 200 hPa.

In this study, the model is used in its tracer version and there is no atmospheric chemistry. We choose to release passive gaseous tracers in the simulation, because we want to analyze only their transport (no chemistry, no deposition) caused by the
25  monsoonal flow. Since we want to distinguish the relative contribution of  several coastal cities to the pollution further inland, we designed a first experiment for which we impose  the tracer emissions at specific urbanized locations: Abidjan (Ivory Coast), Accra (Ghana), Lomé (Togo), Cotonou (Benin), Lagos (Nigeria) (*cf.* Table 1 for coordinates). Specific tracers are emitted for a given city in order to distinguish their relative contributions  at inland locations. Thus the tracer emissions occur in a single grid cell corresponding to the center of each city.

30  The  tracer emissions are constant and continuous during the modeled period ( 1–7 July). This allows quantifying the variability due to the meteorology only.  Emissions are released at the lowest level of the model (below 10 m altitude)  and are proportional to the population of each city, that approach has also been used by Flamant et al. (2018a). We defined an arbitrary emission for one million inhabitants. Then we multiply

[revised manuscript text omitted]

We now focus on the temporal variability reproduced by the tracer experiment at Savè (light green dot in Figure 7). The tracer concentration of the five cities has been interpolated to Savè coordinates (Figure 8-top). The tracer emissions started on 1 July at 00:00 UTC. The first tracer plume that reaches Savè  is that from Cotonou in the evening. We can see that tracers emitted from Cotonou reach Savè every day in the evening, typically at around 19:00 UTC. In the morning, the Lomé pollution plume reaches Savè, while the Accra pollution plume reaches Savè in the afternoon. There is a short period when hourly concentrations are at a maximum every day, and this peak is associated with the arrival of the Cotonou plume in the evening. This pattern is seen repeatedly over the entire  1–7 July period. The model clearly predicts

identified periods when Savè is under the influence of different cities, which implies that these periods correspond to pollution plumes characterized by different chemical ages.

From 5 to 6 July, the contribution of Cotonou decreases and the Accra and Lomé contributions increase, which suggests a modification of wind patterns.  From midnight to the end of the night, there is no city plume reaching Savè. However, we have seen in Section 4.1 that a high CO concentration persists during the night.

The average diurnal cycle of tracers is presented with the contribution of each city (Figure 8-bottom). It confirms that there are distinct periods when Savè is under the successive influences of Lomé in the morning (06:00 to 12:00 UTC), of Accra in the afternoon (12:00 to 18:00 UTC), and of Cotonou in the evening (18:00 to 01:00 UTC). Lagos tracers do not reach Savè because of the southwesterly monsoon flow.

The observed increases in surface concentration may be explained by several processes: long-range transport of the URB plume and/or a local collapse of the nocturnal boundary layer, quickly concentrating locally emitted pollutants. In our case, the dominant effect is the long-range transport, which is mainly associated with the transport of the Cotonou plume. Moreover, during the entire campaign, we did not note any increase of NO at night (Figure AA4 and Figure AA5). These results suggest that the Cotonou plume is affecting Savè during a short period with a maximum between 21:00 and 22:00 UTC (about 2 times greater than the peak magnitude due to Lomé). This is in agreement with observations of CO and $NO_x$ concentrations (*cf.* Section 4.1). We now need to investigate the diurnal cycle of pollutant transport from coastal cities.

**4.3 Mixing and transport of urban anthropogenic and biomass burning**

In this section, results of Tracer Experiment 2 (*cf.* Table 3) are discussed. We present the spatial patterns of URB and BB tracer concentrations averaged over the three layers described in Section 3.2 and then we analyze the vertical structure of these two types of pollution. We focus on 5 July when the BB layer reaches the Guinea Coast (*cf.* Section 3.1). The first layer height is 300 m, which is roughly the minimum PBL top height at night.

In order to analyze the interactions between URB emissions and the BB layer, we  use the information on the spatial characteristics and vertical extent of BB the layer derived from MODIS and CALIPSO observations (Section 3.1) by releasing passive gaseous tracers  with a spatial horizontal extent from 1°W to 2°E at 4.5°N  and with an altitude of ≈ 1.5 km amsl from 5 July at 00:00 UTC to 6 July at 23:00 UTC (*cf.*  Table 3). Although a non-negligible part of BB is transported in the marine PBL, measurements performed during the DACCIWA campaign have confirmed that the BB layer altitude over the ocean is mostly between 1 and 3 km amsl  (Reeves et al., 2010). URB tracers are not separated by city in this experiment. The threshold of URB tracer concentration for the isocontour presented on the maps is Iso2 = Iso1 × 5 (Figure 9).

[revised manuscript text omitted]

 From the wind patterns, we note that the coastal front is present from 09:00 to 15:00 UTC from the coast up to
10   ≈ 7°N.  It leads to the accumulation of URB pollutants. This period is referred as 'Daytime drying' by Deetz et al. (2018). From 16:00 to 02:00 UTC (*respectively* from 03:00 to 08:00 UTC), the meridional wind increases (resp. decreases) in the PBL and URB pollutants are mostly transported northward within the PBL. This period is referred to 'Atlantic inflow' (*resp.* 'Moist morning') by Deetz et al. (2018).

During the 'Daytime drying' period, we notice that URB and BB tracers accumulate along the coastline in the PBL (on 5 and
15   6 July from 11:00 to 15:00 UTC). When dry convection stops (at 16:00 UTC), wind speed quickly increases with a stronger northward component. BB and URB tracers are simultaneously transported northward from the coast. From 16:00 to 02:00 UTC, the front moves towards the north, and the mixture of BB and URB tracers is advected accordingly. A similar diurnal evolution of BB and URB transport is simulated on 5 and 6 July.

The timing of the coastal front propagation in our simulation is in agreement with Adler et al. (2017) and Deetz et al. (2018),
20   who have shown the same regular occurrence of a coastal front that develops during daytime and propagates inland in the evening. After the frontal passage, there is the establishment of the NLLJ (with a jet axis around 250 m amsl), which is also reproduced in our simulation (with an over-estimation of jet axis altitude of about 150 m).

**5   Conclusions**

In this study, several observational datasets together with high-resolution model simulation are used to analyze the diurnal cycle
25   of atmospheric pollution transport over SWA. We focus on two distinct pollution sources, urban anthropogenic and biomass burning pollutants (URB and BB respectively), in order to understand their mixing and their advection inland.

We first studied the dynamics and thermodynamics in the  lowermost troposphere over SWA using aircraft, radio-sounding and ground-based measurements made during part of the DACCIWA field campaign (from 1 to 7 July 2016).
30

The second part of the study uses high-resolution numerical tracer experiments. We analyzed pollution transport from the main urban emission centres of the  Guinea Coast (Abidjan, Accra, Lomé, Cotonou and Lagos) at the super-site of Savè in order to assess the impact on the air quality of remote inland cities characterized by low local emissions.

Observations at Savè (185 km to the north of Cotonou) show that there is a clear diurnal cycle of $NO_2$ and CO, with a maximum occurring every day between 18:00 and 22:00 UTC, suggesting URB transport from remote emission sites. From the tracer experiments, we demonstrated that there are clear and successive periods of the day when air quality in Savè is affected by different city plumes. Precisely, the contribution of tracers released from Lomé is greater than 50 % (of the total amount of tracers) between 02:00 to 12:00 UTC, while from 12:00 to 18:00 UTC tracers released from Accra constitute the main contributor. Then, during 3 hours (from 20:00 to 22:00 UTC), tracers released from Cotonou reach Savè leading to a contribution greater than 80 %, while it is lower than 10 % during the other 15 hours (from 03:00 to 18:00 UTC). Over the period, tracers released from Cotonou represent a contribution of 40 %, from Lomé of 36 % and from Accra of 23 %. Our results suggest that the successive periods affected by different city plumes are characterized by different chemical ages.

~~To assess the impact of BB on the air quality in Savè, we added a BB layer in the tracer experiments based on satellite observations from MODIS ad CALIOP. The experiment suggest that URB and BB (transported from Central Africa) are mixed along the coastline during the day, whereas at night, URB plumes are transported within the shallow PBL (below about 300 m amsl) and the BB layer is mostly transported between the PBL top and 2 km amsl. The mixture of both URB and BB accumulated over coastal areas is transported northward in the surface layer from 16:00 UTC onward and reaches Savè (185 km to the north) at 21:00 UTC.Our results shows that the coastal front is associated with URB and BB accumulation from 11:00 to 16:00 UTC, and its northward moving is associated with the mixture transport of both pollutants, reaching 8°N at 21:00 UTC.~~

The WRF simulation reproduces a diurnal cycle of the wind over SWA in agreement with Adler et al. (2017) and Deetz et al. (2018). Indeed, the structure of the wind changes from the morning to the evening. When the shallow dry convection over the land is well developed, wind speed in the PBL reaches a minimum. A coastal front develops during the day (from 09:00 to 15:00 UTC) and when it ceases, wind speed quickly increases with a stronger northward component.  There is a specific time of the day (16:00 to 02:00 UTC) for the transport of pollutants from the coast toward the north, which affects inland air quality over SWA, human health as well as radiative transfer and the diurnal cycle of low-level clouds.

Our results based on modeling experiments suggest that URB and BB (transported from Central Africa) are accumulated and mixed along the coastline during the day from 09:00 to 16:00 UTC, whereas at night, URB plumes are transported within the  shallow PBL (below about

300 m amsl) and the BB layer is mostly transported between the PBL top and 2 km amsl. The mixture of both URB and BB accumulated over coastal areas is transported northward in the surface layer from 16:00 UTC onward and reaches Savè at 21:00 UTC.

Over SWA, both wind and URB emissions have a diurnal cycle. The strength of numerical tracer experiments is to enable the dichotomy between the variability linked to the meteorology and the emissions by imposing a constant emission (*i.e.* we do not account for any diurnal cycle). In this article, only the observed variability linked to the meteorology is analyzed, we demonstrated that they are clear periods of the day when Savè is impacted by pollution plumes from different cities. In future research, integrated analyses should be conducted to characterize both the URB plumes and the BB layer in terms of composition, gaseous and particulate phase, oxidation of the organic components, and spatio-temporal variability. The DACCIWA campaign provides unique and valuable observations that will allow the investigation of the perspectives opened by this article based on tracer experiments.

[revised manuscript text omitted]

Sayer, a. M., Hsu, N. C., Bettenhausen, C., and Jeong, M. J.: Validation and uncertainty estimates for MODIS Collection 6 "deep Blue" aerosol data, Journal of Geophysical Research: Atmospheres, 118, 7864–7872, https://doi.org/10.1002/jgrd.50600, 2013.

Sayer, A. M., Munchak, L. A., Hsu, N. C., Levy, R. C., Bettenhausen, C., and Jeong, M. J.: MODIS Collection 6 aerosol products: Comparison between Aqua's e-Deep Blue, Dark Target, and "merged" data sets, and usage recommendations, Journal of Geophysical Research: Atmospheres, 119, 13,965–13,989, https://doi.org/10.1002/2014JD022453, 2014.

Schrage, J. M. and Fink, A. H.: Nocturnal Continental Low-Level Stratus over Tropical West Africa: Observations and Possible Mechanisms Controlling Its Onset, Monthly Weather Review, 140, 1794–1809, https://doi.org/10.1175/MWR-D-11-00172.1, 2012.

Schuster, R., Fink, A. H., and Knippertz, P.: Formation and Maintenance of Nocturnal Low-Level Stratus over the Southern West African Monsoon Region during AMMA 2006, Journal of the Atmospheric Sciences, 70, 2337–2355, https://doi.org/10.1175/JAS-D-12-0241.1,

5   2013.

Skamarock, W. C. and Klemp, J. B.: A time-split nonhydrostatic atmospheric model for weather research and forecasting applications, Journal of Computational Physics, 227, 3465–3485, https://doi.org/10.1016/j.jcp.2007.01.037, 2008.

Sobhani, N., Kulkarni, S., and Carmichael, G. R.: Source Sector and Region Contributions to Black Carbon and PM2.5 in the Arctic, Atmospheric Chemistry and Physics Discussions, pp. 1–43, https://doi.org/10.5194/acp-2018-65, 2018.

10  United Nations report: World urbanization prospects: the 2011 revision, Population Division, Department of Economic and Social Affairs, United Nations Secretariat, http://www.un.org/en/development/desa/population/publications/pdf/urbanization/WUP2011_Report. pdf,[Accessed:11/06/2018], 2011.

van Leer, B.: Towards the ultimate conservative difference scheme. V. A second-order sequel to Godunov's method, Journal of Computational Physics, 32, 101–136, https://doi.org/10.1016/0021-9991(79)90145-1, 1979.

15  von Storch, H., Langenberg, H., and Feser, F.: A Spectral Nudging Technique for Dynamical Downscaling Purposes, Monthly Weather Review, 128, 3664–3673, https://doi.org/10.1175/1520-0493(2000)128<3664:ASNTFD>2.0.CO;2, 2000.

Winker, D. M., Vaughan, M. A., Omar, A., Hu, Y., Powell, K. A., Liu, Z., Hunt, W. H., and Young, S. A.: Overview of the CALIPSO Mission and CALIOP Data Processing Algorithms, Journal of Atmospheric and Oceanic Technology, 26, 2310–2323, https://doi.org/10.1175/2009JTECHA1281.1, 2009.

20  Winker, D. M., Tackett, J. L., Getzewich, B. J., Liu, Z., Vaughan, M. A., and Rogers, R. R.: The global 3-D distribution of tropospheric aerosols as characterized by CALIOP, Atmospheric Chemistry and Physics, 13, 3345–3361, https://doi.org/10.5194/acp-13-3345-2013, 2013.

Yang, Y., Wang, H., Smith, S. J., Ma, P.-L., and Rasch, P. J.: Source attribution of black carbon and its direct radiative forcing in China, Atmospheric Chemistry and Physics, 17, 4319–4336, https://doi.org/10.5194/acp-17-4319-2017, 2017.

25  Zuidema, P., Redemann, J., Haywood, J., Wood, R., Piketh, S., Hipondoka, M., and Formenti, P.: Smoke and Clouds above the Southeast Atlantic: Upcoming Field Campaigns Probe Absorbing Aerosol's Impact on Climate, Bulletin of the American Meteorological Society, 97, 1131–1135, https://doi.org/10.1175/BAMS-D-15-00082.1, 2016.

**Appendix A:  Supplemental Material**

[Figure]

**Figure A1.**  _MODIS AOD daily average of two products acquired by Aqua and Terra (the combined Dark-Target and Deep-Blue MYD08-D3 and MOD08-D3 products respectively) over the_  _1–7 July 2016 period (top-left) and over each day of the period. Data excluded by the cloud screening process are in gray. The modeling domain is presented by the black square._

[Figure]

**Figure A2.** *Space Lidar CALIPSO data onboard the CALIOP platform on 5 July 2016 with the orbit track location, the vertical profile of the*
**34**
*total attenuated backscatter, the vertical feature mask and the aerosol subtype. The modeling domain is the red box. (data are available on www-calipso.larc.nasa.gov/products/lidar/browse_images).*

[Figure]

**Figure A3.** *Wind speed (top-left panel) and direction (top-right panel) hourly averages and standard deviations on 5 July 2016 for all aircraft measurements (dots and errorbars) acquired in a box (latitude from 6.6°N to 7.8°N, in longitudes from 1.5°E to 2.2°E, in altitude from 300 m to 1000 m). The horizontal extent of the box and the aircraft tracks are displayed on the map (bottom panel). Modeled value averages of the whole box are presented by the red line (and standard deviation by red shading). Blue dots correspond to ATR-42 measurements, violet to Falcon and yellow to Twin Otter.*

[Figure]

**Figure A4.** *Time series of (top) carbon monoxide (CO), nitrogen dioxide (NO₂) and nitrogen monoxide (NO) hourly concentrations (in ppb) observed at Savè (Benin) over the entire campaign period (25 June to 15 July 2016), and (bottom) of the ratios: CO/NOₓ (in black) and NO₂/NO (in red). The studied period ( 1–7 July 2016) is defined by the two green dashed vertical lines.*

[Figure]

**Figure A5.** *Hourly diurnal cycles of (top) carbon monoxide (CO in black) and (bottom) nitrogen dioxide (NO₂ in blue) and nitrogen monoxide (NO in red) concentrations (in ppb) observed at Savè (Benin). Means of each hour are presented by the lines over the entire campaign period (25 June to 15 July 2016). The upper and lower shading limits correspond to the hourly ranges (maximum and minimum of each hour over the period).*